# Sim2SG: Sim-to-Real Scene Graph Generation for Transfer Learning

## Abstract

Scene graph (SG) generation has been gaining a lot of traction recently. Current SG generation techniques, however, rely on the availability of expensive and limited number of labeled datasets. Synthetic data offers a viable alternative as labels are essentially free. However, neural network models trained on synthetic data, do not perform well on real data because of the domain gap. To overcome this challenge, we propose Sim2SG, a scalable technique for sim-to-real transfer for scene graph generation. Sim2SG addresses the domain gap by decomposing it into appearance, label and prediction discrepancies between the two domains. We handle these discrepancies by introducing pseudo statistic based self-learning and adversarial techniques. Sim2SG does not require costly supervision from the real-world dataset. Our experiments demonstrate significant improvements over baselines in reducing the domain gap both qualitatively and quantitatively. We validate our approach on toy simulators, as well as realistic simulators evaluated on real-world data.

## 1 Introduction

Scene Graphs (SGs) in both computer vision and computer graphics are an interpretable and structural representation of scenes. A scene graph summarizes entities in the scene and plausible relationships among them. SGs (Dai et al., 2017; Herzig et al., 2018; Li et al., 2017; Newell & Deng, 2017; Xu et al., 2017; Yang et al., 2018; Zellers et al., 2018) are a manifestation of vision as inverse graphics. They have found a variety of applications such as image captioning, visual question answering, high level reasoning tasks, image retrieval, image generation, etc. However, most prior work on SG generation relies on the availability of expensive and limited number of labeled datasets such as Visual Genome (Krishna et al., 2017) and Visual Relationship Dataset (VRD) (Lu et al., 2016).

One of the main limitations in machine learning applications is the general lack of sufficient labeled data for supervised learning tasks. Synthetic data is a viable alternative to this problem since annotations are essentially free. Synthetic data has been used for a variety of tasks such as image classification, object detection, semantic segmentation, optical flow modeling, 3D keypoint extraction, object pose estimation, 3D reconstruction, etc. (Borrego et al., 2018; Butler et al., 2012; Dosovitskiy et al., 2015; McCormac et al., 2016; Mueller et al., 2017; Richter et al., 2016; Ros et al., 2016; Suwajanakorn et al., 2018; Tremblay et al., 2018; Tsirikoglou et al., 2017). It has also been shown to be effective in initializing task networks (Prakash et al., 2019) and for data augmentation. However, the use of synthetic data for SG generation and visual relationships is yet to be explored.

One crucial issue with training on a labeled source domain (synthetic data) and evaluating on an unlabeled target domain (real data) is the performance gap known as *domain gap* (Torralba & Efros, 2011). This gap is due to the difference of data distribution between the source and target domains. Kar et al. (2019) argue that domain gap can be divided into *appearance* and *content* gap. The appearance gap can be addressed by making scenes photo-realistic (McCormac et al., 2016; Wrenninge & Unger, 2018), by using image translations (Hoffman et al., 2018; Huang et al., 2018; Zhu et al., 2017), by feature alignment (Chang et al., 2019; Chen et al., 2018; Li et al., 2019; Luo et al., 2019; Saito et al., 2019; Sun et al., 2019), or by learning robust representations based on domain randomization (Prakash et al., 2019; Tobin et al., 2017). There are also studies that address the content gap for image classification (Azizzadenesheli et al., 2019; Lipton et al., 2018; Tan et al., 2019). We present a thorough investigation of the domain gap between source and target domains. We assume a gap in both appearance and content, expand those gaps into different sub-components and provide a way to address them. We primarily apply our method to reduce the domain gap for

SG generation. Nonetheless, our techniques can also be applied to other vision tasks such as image classification, image segmentation and object detection among others.

We propose Sim2SG (Simulation to Scene Graph); a model that learns sim-to-real scene graph generation leveraging labeled synthetic data and unlabeled real data. Extending the formulation in (Wu et al., 2019), Sim2SG addresses the *domain gap* by bounding the task error (where the task is scene graph generation) on real data through appearance, prediction, label (ground truth) discrepancies between the two domains and task error on synthetic data. Our work differs from (Wu et al., 2019) as they do not provide a way to address the content gap, and their risk discrepancy is intractable. To the best of our knowledge, Sim2SG is the first work to introduce a tractable error bound on the content component of the domain gap.

We minimize the appearance and prediction discrepancies by aligning the corresponding latent and output distributions via Gradient Reversal Layers (Ganin et al., 2017). We address discrepancy in the label using principles of *self-learning* (Zou et al., 2018). However, self-learning based on pseudo labels often suffer from the inaccurately generated labels (e.g. predicted bounding boxes are ill-categorized or imprecise, hence, the model will regress on the wrong objects) (Zheng & Yang, 2020; Kim et al., 2019). Therefore, we instead propose to collect a higher level statistic (e.g. list of objects and their type, position and relationships for placement), that we call *pseudo-statistics*, from target data and leverage the synthetic data generator to produce valid objects with their precise labels (e.g. bounding boxes). We experimentally demonstrate our method in three distinct environments–all synthetic CLEVR (Johnson et al., 2017), more realistic Dining-Sim and Drive-Sim with a driving simulator evaluated on KITTI (Geiger et al., 2012). We almost close the domain gap in the Clevr environment and we show significant improvements over respective baselines in Dining-Sim and Drive-Sim. Through ablations, we validate our assumptions about appearance and content gaps. Sim2SG differs from other unsupervised domain adaptation methods (Chen et al., 2018; Xu et al., 2020; Li et al., 2020) as it can modify the source distribution (via self-learning based on pseudo-statistics to align with the target distribution) with access to a synthetic data generator. We also outperform these domain adaptation baselines (Chen et al., 2018; Xu et al., 2020; Li et al., 2020) as shown in Section 4.3.

**Contributions:** Our contributions are three-fold: In terms of methodology, to the best of our knowledge, (1) We are the first to propose sim-to-real transfer learning for scene graph generation. We do not require costly supervision from the target real-world dataset. (2) We study domain gap from synthetic to real data in detail, provide a tractable error bound on the content component of the gap and propose a novel pipeline including *pseudo statistics* to fully handle the gap. Experimentally, (3) we show that Sim2SG can learn SG generation and obtains significant improvements over baselines in all three scenarios - Clevr, Dining-Sim and Drive-Sim. We also present ablations to illustrate the effectiveness of our technique.

## 2 PROPOSED METHOD: SIM2SG

Our proposed Sim2SG pipeline is illustrated in Figure 1. We first describe how we generate scene graphs in Section 2.1. When we naïvely train on a source distribution (synthetic data) and evaluate on a target distribution (real data), we have a *domain gap* (Torralba & Efros, 2011). We study it in more detail in Section 2.2 and propose methods to address it.

### 2.1 SCENE GRAPHS

This section describes scene graphs (SGs) and how we train the SG predictor module using labels from the source domain.

**Notation:** We represent a scene graph of a given image $I$ as a graph $G$ with nodes $o$ and edges $r$. Each node is a tuple $o_i = \langle b_i, c_i \rangle$ of bounding box $b_i = \{xmin_i, ymin_i, w_i, h_i\}$ and category $c_i$. Relationships $r$ are a triplet of $\langle o_i, p, o_j \rangle$ where $p$ is a predicate. SG prediction has two key components: feature extractor $\phi$ and graph predictor $h$. $\phi$ maps input space $x$ to a latent space $z$ and $h$ maps from latent space $z$ to output space $y$. The predicted SG is $G = h(\phi(x))$.

We use Resnet 101 (He et al., 2016) to implement $\phi$ and GraphRCNN (Yang et al., 2018) architecture to implement $h$. We train the networks $\phi$ and $h$ using the following task loss (Yang et al., 2018): cross entropy loss for object classification & relationship classification and $\ell_1$ loss for bounding boxes.

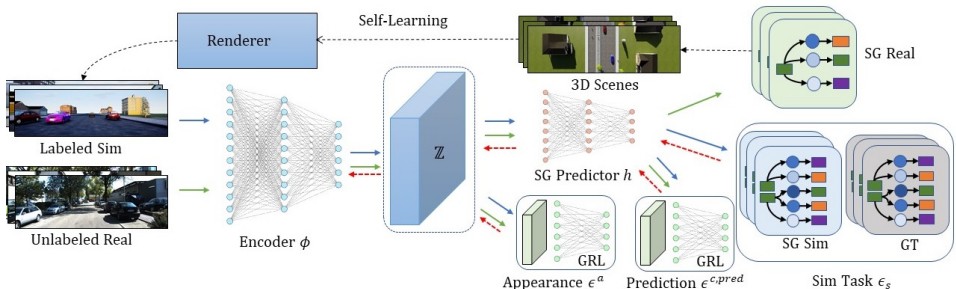

Figure 1: Overview of Sim2SG. We first map the labeled synthetic and unlabeled real data to a shared representation Z using encoder $\phi$. Then we train scene graph prediction network $h$ on Z using synthetic data. We align label discrepancies (content gap) between the two domains using pseudo statistics based self-learning. We further bridge the domain gap by aligning features in the representation space Z (appearance gap) and output space (content gap) using Gradient Reversal Layer (GRL) and domain discriminator (Ganin et al., 2017). Blue, green and red arrows indicate the flow of synthetic data, real data and back propagation, respectively.

Our framework Sim2SG is illustrated in Figure 1. It is worth noting that we predict the relationships among objects in the scene, but not their attributes like (Yang et al., 2018).

## 2.2  DOMAIN GAP

We now study the domain gap between source and target domains and formulate the SG generation task error on real domain as a function of the task error on synthetic domain and appearance & content gap between the domains. We then propose methods to address each subcomponent of the gap.

**Notation and assumptions:** We have two domains: synthetic $\langle x_s, y_s \rangle \sim p(x, y)$ and real domain $\langle x_r, y_r \rangle \sim q(x, y)$. Note that $x_s$ or $x_r$ is the input (image) and $y_s$ or $y_r$ is the output (SG) as introduced in Section 2.1 It is also worth noting that $y_r$ is not known. We assume that synthetic and real domains contain the same categories of objects. Both domains also share similar scenarios (*e.g.* both have driving scenes). However, joint distributions of scenes (images) are different in both domains (*i.e.* $p(x) \neq q(x)$). The label (ground truth) distributions are also different in the two domains (*i.e.* $p(y) \neq q(y)$). Using the formulation in (Wu et al., 2019), the task (SG generation) error (i.e. risk) on synthetic domain as a function of the latent space $z$ is given by:

$$\epsilon_s(\phi, h) = \int p(z) e_s dz \qquad (1)$$

where $e_s$ is the risk defined as $e_s = |p(y|z) - y_s|$. $p(z)$ is the distribution of features, $p(y)$ is the distribution of labels and $p(y|z)$ is the output distribution. The bound on task error in real domain $\epsilon_r(\phi, h)$ from Wu et al. (2019) is a function of three terms which are: (1) task error on the synthetic domain $\epsilon_s(\phi, h)$, (2) risk discrepancy between the domains $\epsilon^c(\phi, h)$ and (3) feature discrepancy between the two domains $\epsilon^a(\phi, h)$. Our goal is to minimize the task error on the real domain:

$$\epsilon_r(\phi, h) = \int q(z) e_r dz = \int q(z) e_r dz + \int p(z) e_s dz - \int p(z) e_s dz + \int q(z) e_s dz - \int q(z) e_s dz$$

$$= \int p(z) e_s dz + \int q(z)(e_r - e_s) dz \int (q(z) - p(z)) e_s dz$$

$$= \epsilon_s(\phi, h) + \qquad \epsilon^c(\phi, h) \qquad + \qquad \epsilon^a(\phi, h)$$

$$(2)$$

We drop the terms $\phi, h$ from now on. We would need to minimize $\epsilon_s$, $\epsilon^a$ and $\epsilon^c$ to reduce the task error on the real domain $\epsilon_r \geq 0$. If the error $\epsilon_r$ reduces to zero on the target domain, we have *closed* the domain gap. Since we have access to the label $y_s$, we can minimize the task error on synthetic domain $\epsilon_s$ as described in Section 2.1. However, we do not have access to the label distribution $q(y)$ for the real domain. This makes the risk discrepancy $\epsilon^c$ intractable. We split this discrepancy into tractable components and address them in Section 2.2.1. We call the discrepancy *content gap* based on our

---

**Algorithm 1** Pseudocode for Sim2SG training

---

1: **Given:** $\phi_\theta, h_\theta$                        ▷ Encoder, Scene Graph predictor
2: **Given:** $X_s, Y_s, R, X_r$           ▷ Synthetic Images, Synthetic Labels, Data Generator, Real Images
3: **Hyperparameters:** $E, I$                           ▷ Epochs, Iterations
4: **for** $e \in E$ **do**                                     ▷ Training
5:      $loss = 0;$
6:      **for** $i \in I_m$ **do**
7:          $loss \mathrel{+}= \epsilon^a(\phi(x_s), \phi(x_r))$                    ▷ Appearance gap
8:          $loss \mathrel{+}= \epsilon^{c,pred}(h(\phi(x_s)), h(\phi(x_r)))$          ▷ Content prediction gap
9:          $loss \mathrel{+}= task(x_s, y_s)$                ▷ Scene Graph generation task loss
10:          $\phi_\theta, h_\theta = \text{optimize}(\phi_\theta, h_\theta, loss);$        ▷ Network parameters $\theta$ ; SGD step
11:      **end for**
12:      $ps = \{h(\phi(x_r)) : x_r \in X_r\}$           ▷ extracting pseudo statistics from predictions
13:      $X_s, Y_s = R(ps)$           ▷ generating synthetic data with labels aligned to real data
14: **end for**

---

ablations in Section 4.1. Recent work shows that features frequently account for appearance (Geirhos et al., 2019), and therefore we call the feature discrepancy $\epsilon^a$ as *appearance gap* and empirically show through ablations in Section 4.1 to be the case. We address this discrepancy in Section 2.2.2. The pseudo-code of our proposed method is illustrated in Algorithm 1.

### 2.2.1 CONTENT GAP: $\epsilon^c$

Content gap refers to discrepancies between the two domains including the difference in distribution of the number of objects and their class, placement, pose and scale. This affects the position, dimension and the type of the labels leading to the problem of label shift (i.e. discrepancy in the ground truth distribution). However, minimizing $\epsilon^c$ in the current form is not tractable. Since the task error (risk) $\epsilon_r$ is non-negative and both $\epsilon_r$ and $e_r$ are lower bounded by zero, we also assume a lower bound of zero for $\epsilon^c$. Hence, we approximate the $(e_r$ - $e_s)$ as $((q(y|z) - p(y|z)) + (y_s - y_r))$. We show this in equation 3. We assume $y_s \equiv y_r$ and $q(y|z) \equiv p(y|z)$ as a sufficient condition for $((q(y|z) - p(y|z)) + (y_s - y_r))$ to be zero and to avoid any degenerate solutions. Based on these assumptions, we also empirically demonstrate in Section 4 that the label discrepancy $\epsilon^{c,label}$ and the prediction discrepancy $\epsilon^{c,pred}$ help reduce the domain gap.

$$\begin{aligned}
\arg\min_\theta \epsilon^c &= \arg\min_\theta \int q(z)(e_r - e_s)dz \\
&\simeq \arg\min_\theta \int q(z)(y_s - y_r)dz + \int q(z)(q(y|z) - p(y|z))dz \\
&= \arg\min_\theta \qquad \epsilon^{c,label} \quad + \quad \epsilon^{c,pred}
\end{aligned} \tag{3}$$

**Label discrepancy and Pseudo-Statistics:** Minimizing $\epsilon^{c,label}$ is challenging because we do not have access to the label $(y_r)$ of the target (real) domain. We then propose to get an estimate of $y_r$ through principles of self-learning (Zou et al., 2018) based on minimum reliable statistic (pseudo-statistic) of target data and generate aligned synthetic data $(y_s)$ to narrow the gap. We generate SGs for all input images $(x_r)$ of the target domain and derive the *pseudo statistic ps* from each SG by retaining the minimum information needed for 3D representation (e.g. we discard the entire bounding box and keep the centroid position). The statistic $ps$ is a list of objects with each object's type, centroid location and relationship with others. Using either known or assumed camera intrinsic (e.g. car dash cam), we map $ps$ to a full 3D scene. Some unknown parameters (e.g. texture or pose) and context (ground, sky, light) are randomized as done in (Prakash et al., 2019). We use a synthetic data generator to render those scenes as shown in Figure 11. More details on scene generation for different environments are in Sections A.2.1, A.2.2 and A.2.3 of Appendix. This is also analogous to an Expectation Maximization algorithm where we compute pseudo-statistics and generate aligned synthetic data (E-step) and then use it to train the Sim2SG model (M-step).

Contrarily, self-learning based on pseudo labels (Zou et al., 2018) generated from and trained on real data often suffer from poorly generated labels (Zheng & Yang, 2020; Kim et al., 2019). For instance, training SG generation model on ill-categorized or imprecise bounding boxes can lead to poor detection results. Pseudo-Statistics (described previously) integrated with a synthetic data generator (renderer) will still produce a valid scene for those ill-detected objects (false positives) with

precise labels (bounding boxes). We show through quantitative experiments that our method performs better than pseudo label based self-learning in Section 4.3. Please note that our method is orthogonal to pseudo labels based self-learning and the latter can potentially be applied in conjunction.

**Prediction discrepancy:** The output of the scene graph generation model should be the same for same categories in different domains. To address the prediction discrepancy $\epsilon^{c,pred}$, we propose to align output distribution for latent code belonging to the same visual category. To this end, we align the output distributions $p(y|z)$ and $q(y|z)$ using a GRL based technique (Ganin et al., 2017) that we will discuss in Section 2.2.2.

### 2.2.2 APPEARANCE GAP: $\epsilon^a$

Appearance gap is the discrepancy in the appearance of the two domains. This includes differences in texture, color, light, reflectance, etc. of objects in the scene. To address the appearance gap, we want to avoid photo-realism in synthetic data as it requires high quality assets and a tremendous amount of effort from artists (McCormac et al., 2016; Wrenninge & Unger, 2018). Therefore, we propose to learn an appearance invariant representation $z$ so we can avoid the appearance bias in the model. However, the features $z : \phi(x)$ have both content and appearance components and aligning them may be detrimental (Saito et al., 2019; Wu et al., 2019) as shown in the ablation in Section 4.3. To overcome this challenge, we align the appearance gap $\epsilon^a$ only after we align the content gap $\epsilon^c$ of the two domains using the method described in Section 2.2.1. We minimize this appearance gap $\epsilon^a$ by aligning the feature distributions $p(z)$ and $q(z)$ as follows:

$$\boldsymbol{\theta}^* = \arg\min_{\boldsymbol{\theta}} \int (q(z) - p(z))e_s dz \tag{4}$$

where we exploit the fact that $p(z) \equiv q(z)$ is a sufficient condition for $\epsilon^a$ to be zero (Wu et al., 2019). We use Gradient Reversal Layer (GRL) (Ganin et al., 2017) to align the distributions $p(z)$ and $q(z)$ along with a domain classifier $D$ to classify them. We minimize the $D$'s loss *w.r.t.* its own parameters while maximizing *w.r.t.* the network parameters of $\phi$ ($\theta$). We do so through GRL that acts as an identity function during forward propagation and flips the sign of the gradients during back propagation from $D$ to $\phi$. We provide details of GRL layers in Section A.1

## 3 RELATED WORK

**Scene Graphs:** There has been a plethora of work on Visual Relationships and Scene Graph (SG) generation (Dai et al., 2017; Herzig et al., 2018; Li et al., 2017; Newell & Deng, 2017; Xu et al., 2017; Yang et al., 2018; Zellers et al., 2018). While all this work is fully supervised and some use priors, they still depend on availability of expensive, large scale data (Krishna et al., 2017; Lu et al., 2016). Krishna et al. (2019) propose a way to generate SGs using limited labels and Dornadula et al. (2019) learn to generalize SG prediction to rare categories in a few shot setting. However, they still use either existing object detection models trained on large data or ground truth object labels. We, on the other hand, require no labels whatsoever in the target domain. Alternatively, there are also works on scene generation from SG (Ma et al., 2018; Herzig et al., 2020)

**Synthetic Data** has been used for many tasks including, but not limited to, object detection (Kar et al., 2019; Prakash et al., 2019), semantic segmentation (Richter et al., 2016; Ros et al., 2016; Tsirikoglou et al., 2017), optical flow modeling (Butler et al., 2012; Dosovitskiy et al., 2015), scene flow (Mayer et al., 2016), classification (Borrego et al., 2018), stereo (Qiu & Yuille, 2016; Zhang et al., 2016), 3D keypoint extraction (Suwajanakorn et al., 2018), object pose estimation (Mueller et al., 2017; Tremblay et al., 2018) and 3D reconstruction (McCormac et al., 2016). However, to the best of our knowledge, synthetic data has not been applied to scene graph generation.

**Domain Gap** is the performance gap when the network is trained on a synthetic domain and evaluated on real data. Kar et al. (2019) argue that domain gap has two components which are appearance and content. We work with the same assumptions. Most prior work addresses the appearance gap by image translations (Chen et al., 2019; French et al., 2018; Hoffman et al., 2018; Huang et al., 2018; Li et al., 2018; Zhu et al., 2017), clever feature alignment (Chen et al., 2018; Li et al., 2019; Luo et al., 2019; Saito et al., 2019; Xu et al., 2020; Li et al., 2020) and domain randomization (Prakash et al., 2019; Tobin et al., 2017). There are few works which handle the content gap (Azizzadenesheli et al., 2019; Lipton et al., 2018; Tan et al., 2019; Zhao et al., 2019). More specifically, they address the *label shift* between the two domains. However, they do not exploit the unlabeled images from

Table 1: Quantitative results of Sim2SG when evaluated on the target domain in CLEVR environment.

| Trained on | Cube AP | Cylinder AP | Sphere AP | mAP @0.5 IoU | Recall@20 |
|---|---|---|---|---|---|
| Source only | 0.479 ±0.130 | 0.810 ±0.048 | 0.881 ±0.024 | 0.723 ±0.053 | 0.356 ±0.047 |
| Source + $\epsilon^{c,label}$ | 0.805 ±0.081 | 0.779 ±0.093 | 0.913 ±0.027 | 0.832 ±0.046 | 0.493 ±0.064 |
| Source + $\epsilon^a$ | 0.831 ±0.129 | 0.723 ±0.176 | 0.908 ±0.002 | 0.821 ±0.048 | 0.815 ±0.026 |
| Source + $\epsilon^a$ + $\epsilon^{c,label}$ | **0.903** ±0.009 | **0.827** ±0.098 | **0.944** ±0.034 | **0.892** ±0.024 | **0.888** ±0.018 |

the target domain. We, on the other hand, leverage the images from the target domain to reduce the domain gap further. We are also interested in a scene graph generation task which is more complex than classification. The idea of self-training with *pseudo labels* (Zou et al., 2018) is used in (Li et al., 2019; Tan et al., 2019) to learn models from target distribution. However, the labels predicted by the model on the target are often inaccurate because of domain gap (Zheng & Yang, 2020). We instead rely on pseudo statistics and use a synthetic data generator to produce accurate labels. Similar to us, Chang et al. (2019); Sun et al. (2019) also train their task model on top of domain invariant features for image classification and image segmentation.

## 4 EXPERIMENTS

We evaluate Sim2SG in three different environments with increasing complexity. In each environment we have a fully labeled source domain and unlabeled target domain with labeled test data. We present a simple environment using CLEVR (Johnson et al., 2017) as described in Section 4.1 and a more realistic synthetic environment in Dining-Sim using ShapeNet (Chang et al., 2015) in Section 4.2. We then use an Unreal Engine 4 based driving simulator similar to (Prakash et al., 2019) as the source domain, and real images from KITTI (Geiger et al., 2012) as the target domain in Section 4.3. Using these experiments, we show that Sim2SG learns sim-to-real scene graph (SG) generation, it reduces the domain gap, and our label alignment (Eq. (3)), prediction alignment (Eq. (3)) and appearance alignment (Eq. (4)), work as intended. We compare Sim2SG to several baselines in DriveSim environment (Section 4.3). Our quantitative evaluation metric includes detection mAP (mean average precision) @ 0.5 IoU (Intersection over Union) and relationship triplet recall @20 or @50 (Krishna et al., 2017). Note that relationship triplet recall implicitly includes object detection recall as well (see Appendix). All the mean and standard deviations are based on five runs.

### 4.1 TOY EXPERIMENTS ON THE CLEVR DATASET

The goal of the experiments on the CLEVR environment (Johnson et al., 2017) is to show that Sim2SG can learn sim-to-real scene graph generation on an unlabeled domain while addressing the *domain gap* between source and target domains. Additionally, we show that appearance alignment $\epsilon^a$ and label alignment $\epsilon^{c,label}$ work as intended through ablations. For our experiments, we have 3 classes of objects: cube, sphere and cylinder and 4 kinds of relationships: front, behind, left and right. Our source domain mimics the synthetic environment by having simple texture, different number of objects (4) and closer objects. Nonetheless, objects are placed at random locations in the scene as shown in the top row of Figure 4. The target domain is representative of real data with a different set of colors, number of objects (2 or 3) and objects placed with larger margin. Regardless, the objects can be placed anywhere and have more complex texture by applying a style transfer network to the generated scenes (second row of Figure 4). We use 1000 labeled images of source, 1000 unlabeled images of target for training and 200 labeled images of source and target for evaluation. Details of the environment, training and hyper-parameters are in Section A.2.1 of Appendix.

**Results:** Quantitative evaluation of Sim2SG is reported in Table 1. When we train on the source domain and evaluate on a test set from the source domain itself, we achieve 1.0 mAP @0.5 IoU for detection 0.986 recall@20 for relationship triplets. Hence, the first row shows that there is a domain gap from source to target as seen from the lower mAP and recall values. Second and third rows show how label alignment $\epsilon^{c,label}$ and appearance alignment $\epsilon^a$, drastically **reduce** the domain gap compared to baseline (source only). We see that $\epsilon^{c,label}$ improves detection performance as it is possible that the detection recall is more sensitive to content gap. Nonetheless, $\epsilon^a$ is more effective in improving the relationship triplet recall as we have found it to reduce false positive detections effectively (Figure 6). The domain gap reduces further by combining both $\epsilon^{c,label}$ and $\epsilon^a$ terms (fourth row). Please note that we report saturation performance (details in Appendix). Qualitative improvements of scene graph recall over baseline are shown in first row of Figure 2.

Table 2: Left (resp. right): Source and target domains have different (resp. similar) appearance but similar (resp. different) content distribution. All the evaluations are on the target domain.

| Trained on | mAP @0.5 IoU | Recall@20 | Trained on | mAP @0.5 IoU | Recall@20 |
|---|---|---|---|---|---|
| Source only | 0.675 | 0.339 | Source only | **1.0** | 0.76 |
| Source + $\epsilon^{c,label}$ | 0.923 | 0.646 | Source + $\epsilon^{a}$ | 0.97 | 0.722 |
| Source + $\epsilon^{a}$ | **0.938** | **0.938** | Source + $\epsilon^{c,label}$ | **1.0** | **0.996** |

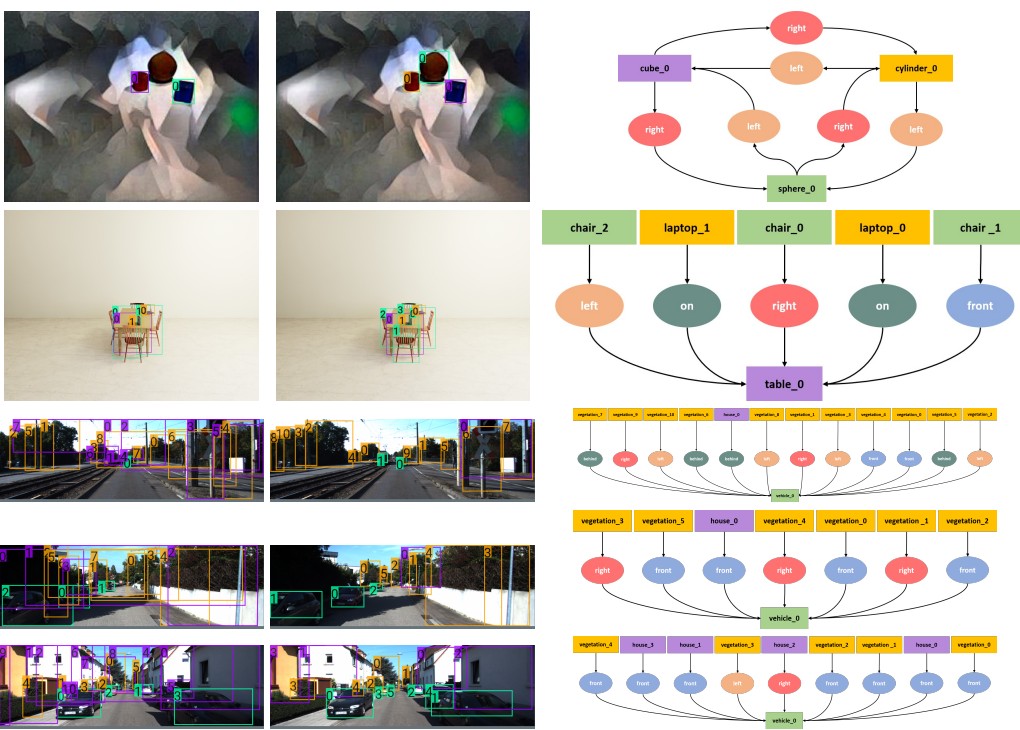

Figure 2: Qualitative results of Sim2SG on the target domain for CLEVR (first row), Dining Sim (second row) and Drive-Sim environments (last three rows). First column shows that the source only baseline fails to either detect objects or have high number of false positives (mislabels) leading to poor scene graph. Our method detects objects better, has way fewer false positives and ultimately generates more accurate scene graphs as shown in second and third column respectively. Objects are color coded. For better visibility, we only show partial scene graph for Drive-Sim.

**Ablations:** We conduct two sets of experiments on the CLEVR dataset and quantitative results are shown in Table 2. The first experiment studies appearance gap: source and target have the same number of objects and range of margin, but they use different color and texture. Additionally, we transform the target by using the style transfer network. We observe that style alignment $\epsilon^{a}$ reduces the domain gap with significant improvement over baseline. However, the label alignment $\epsilon^{c,label}$ fails to have significant improvement on relationship triplet recall. Similarly, the second experiment studies content gap only (i.e. source and target use same color and texture but different number of objects and margin). We observe that label alignment $\epsilon^{c,label}$ closes the domain gap **completely**. However $\epsilon^{a}$ leads to performance degradation. These experiments show our label alignment $\epsilon^{c,label}$ reduces content gap and appearance alignment $\epsilon^{a}$ addresses appearance gap.

## 4.2 EXPERIMENTS ON DINING-SIM

The goal of our experiments on Dining-Sim created from ShapeNet objects (Chang et al., 2015) is to show that Sim2SG works as intended on a more complex dataset where target domain is representative of real data. This dataset has 3 classes of objects – chair, table and laptop. There are 5 kinds of relationships – front, behind, left, right, and on. The source and target domains are illustrated in the top two rows of Figure 7 and details are in Appendix. The results agree with the findings of Section 4.1. We also see domain gap in this environment as evaluation on source domain (1.000

Table 3: Evaluation on KITTI hard when training Drive-Sim synthetic environment. The class specific AP and mAP are reported at 0.5 IoU.

| Trained on | Car | Pedestrian | House | Veg. | mAP | Recall@50 |
|---|---|---|---|---|---|---|
| Source only (Prakash et al., 2019) | 0.382 ±0.029 | 0.168 ±0.017 | 0.211 ±0.023 | 0.174 ±0.010 | 0.234 ±0.006 | 0.070 ±0.007 |
| DA-FasterRCNN (Chen et al., 2018) | 0.424 ±0.028 | 0.170 ±0.024 | 0.200 ±0.041 | 0.169 ±0.014 | 0.241 ±0.014 | 0.074 ±0.015 |
| Meta-Sim (Kar et al., 2019) | 0.413 ±0.009 | 0.197 ±0.027 | 0.236 ±0.009 | 0.164 ±0.023 | 0.253 ±0.003 | 0.075 ±0.005 |
| Self-learning (Zou et al., 2018) | 0.312 ±0.006 | 0.167 ±0.015 | 0.191 ±0.003 | 0.263 ±0.006 | 0.233 ±0.004 | 0.062 ±0.003 |
| GPA (Xu et al., 2020) | 0.174 ±0.040 | 0.011 ±0.016 | 0.106 ±0.031 | 0.059 ±0.027 | 0.087 ±0.020 | 0.015 ±0.005 |
| SAPNet (Li et al., 2020) | 0.362 ±0.054 | 0.085 ±0.051 | 0.116 ±0.021 | 0.067 ±0.022 | 0.157 ±0.024 | – |
| Ours ($\epsilon^{c,label}$) | 0.410 ±0.009 | **0.262** ±0.025 | 0.240 ±0.010 | 0.229 ±0.036 | 0.285 ±0.003 | 0.104 ±0.006 |
| Ours ($\epsilon^{c,label} + \epsilon^a$) | 0.493 ±0.004 | 0.252 ±0.014 | 0.247 ±0.012 | 0.253 ±0.020 | 0.311 ±0.311 | 0.127 ±0.004 |
| Ours ($\epsilon^{c,label} + \epsilon^a + \epsilon^{c,pred}$) | **0.501** ±0.006 | 0.241 ±0.018 | **0.254** ±0.010 | **0.269** ±0.014 | **0.316** ±0.004 | **0.139** ±0.004 |

mAP @0.5 IoU & 0.995 recall@50) is higher than on target domain (0.584 mAP @0.5 IoU & 0.331 recall@50). The oracle performance on target domain stands at 0.904 mAP@0.5 IoU & 0.846 recall@50. Label alignment $\epsilon^{c,label}$ drastically improves performance on target domain (0.713 mAP @0.5 IoU & 0.501 recall@50). $\epsilon^a$ reduces false positives (Figure 10). We achieve the best scene graph recall @50 using a combination of label alignment $\epsilon^{c,label}$, appearance alignment $\epsilon^a$ and prediction alignment $\epsilon^{c,pred}$ (0.729 mAP @0.5 IoU & 0.547 recall@50). Complete results that validate our approach are in Table 4 in Appendix. Qualitative improvements of scene graph generation over baseline are illustrated in the second row of Figure 2.

## 4.3 REAL-WORLD EXPERIMENTS ON DRIVE-SIM

In this section, we validate our approach on a real-world dataset. For synthetic data, we use a simulator similar to (Prakash et al., 2019) with minor simplifications (fixing camera and road spline parameters, exclusion of some objects, see details in Appendix) to make our generation easier. The number of lanes, sidewalk, cars, vegetation, houses, pedestrians; their positions, pose, color, texture; light settings are randomly picked from a set of realistic values (details in Appendix.) akin to (Prakash et al., 2019) as shown in Figure 13. We use four classes: car, pedestrian, vegetation, house and four types of relationships: front, left, right, behind. All our relationships have the car as the subject. For example, 'car behind car', 'vegetation left car', etc. Although we show the 'on' relationship to work in Dining-Sim environment, we found these relationships were always trivial to predict because they are always true: e.g. cars are always 'on' road, pedestrian 'on' sidewalk, etc. Therefore, we did not include them in the experiments. We use KITTI (Geiger et al., 2012) as the target domain. We need a small amount of labels on KITTI for evaluation **only**. Hence, while keeping the existing annotations for cars and pedestrians, we add annotations for vegetation and houses along with relationships among them. We will release these annotations to the community. We use 6000 labeled synthetic images, 6000 unlabeled KITTI images for training, and 1000 labeled synthetic and 550 labeled KITTI images for evaluation. See Appendix for details on KITTI annotation schema, training and hyper-parameters.

**Baselines**: We compare Sim2SG to the randomization based method (Prakash et al., 2019), the method addressing content gap (Kar et al., 2019), self-learning based on pseudo labels (Zou et al., 2018) and domain adaptation methods for object detection (Chen et al., 2018; Xu et al., 2020; Li et al., 2020).

Prakash et al. (2019) use a context based randomization of pose, position and texture of objects of interest(car). Kar et al. (2019) learn the parameters of a renderer to match the target distribution to address the content gap. Unsupervised domain adaptation methods (Chen et al., 2018; Xu et al., 2020; Li et al., 2020) align the features from source and target domain. We discuss self-learning based on pseudo labels (Zou et al., 2018) in Section 2.2.1. We train our baselines on 6000 images, using the hyper parameters provided by the authors. We adapt the baselines to our framework with Resnet101 as backbone. The details can be found in Section A.2.3.

**Results:** We evaluate scene graph generation on three KITTI evaluation modes : easy, moderate and hard based on object size, occlusion and truncation (Appendix). In all three settings, we improve over our baselines. We report the results for KITTI hard in Table 3, the detailed results for KITTI easy and moderate can be found in Table 6. We perform better than Meta-Sim as it cannot align the structure of the scenes (e.g. number of object in a scene) to match the target distribution. Our predicted scene graphs also differ from the intermediate scene graphs in Meta-Sim which lacks the notion of relationships. We outperform other domain adaptation methods (adapted to our framework) for object detection (Chen et al., 2018; Xu et al., 2020; Li et al., 2020) because we believe that feature alignment without good label/content alignment may not be effective. We discussed this

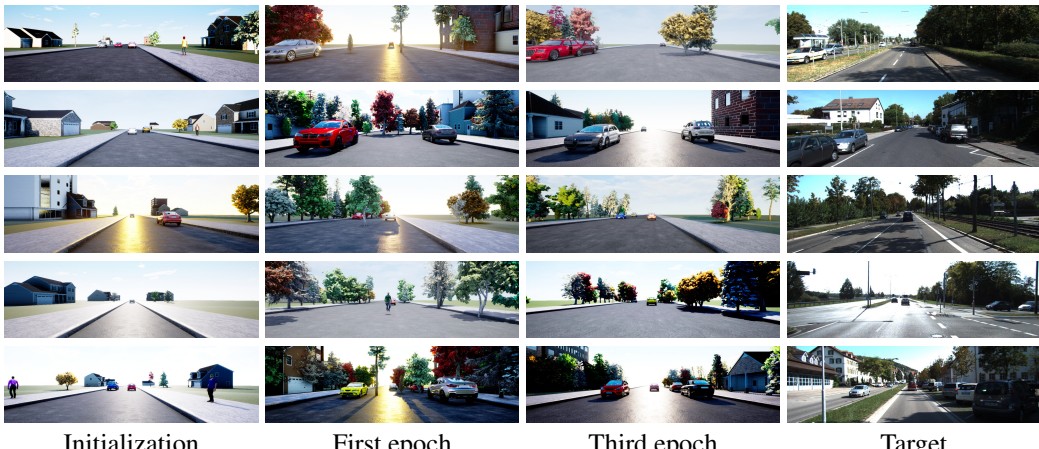

| Initialization | First epoch | Third epoch | Target |

Figure 3: Synthetic data in Drive-Sim environment changing through the training using Label alignment $\epsilon^{c,label}$ . From left to right: synthetic data at initialization (different number, placement of objects w.r.t KITTI), after first epoch (alignment of number and placement of objects, but with noise), after third epoch (better alignment of number and placement of objects w.r.t KITTI) and corresponding KITTI samples)

briefly in Section 2.2.2 and further show an ablation (next paragraph) that Sim2SG also suffers from the same issue. We believe that our label alignment $\epsilon^{c,label}$ can be used in conjunction with other domain adaptation methods. The reason we improve over self-learning based on pseudo labels is also discussed in Section 2.2.1. The last three rows of Table 3 show that most improvements come from label alignment $\epsilon^{c,label}$ and appearance alignment $\epsilon^a$ and the combination of $\epsilon^{c,label}$, $\epsilon^a$ and $\epsilon^{c,pred}$ achieves the best relationship triplet recall. We notice that the AP of the pedestrian category does not improve with $\epsilon^a$ and $\epsilon^{c,pred}$. The reason might be that pedestrians are under-represented, small and hard to detect class in KITTI. Sim2SG can align the label distribution but cannot address the class imbalance in the target domain. The qualitative results are shown in last three rows of Figure 2. We see that Sim2SG significantly improves on both false positives and recall of objects. As a result, it generates more accurate scene graphs. This is because label alignment $\epsilon^{c,label}$ generates nicely aligned data (Fig. 11) and appearance alignment $\epsilon^a$ reduces false positives (see Fig. 2). Figure 3 qualitatively shows how synthetic data is adjusted over the duration of training towards more label alignment w.r.t KITTI using $\epsilon^{c,label}$.

**Ablations:** As briefly discussed in Section 2.2.2, we run the label alignment $\epsilon^{c,label}$ before appearance alignment $\epsilon^a$ and prediction alignment $\epsilon^{c,pred}$ to address the fact that feature alignment can be detrimental if the content of both domains are not aligned. We indeed found that our performance drops significantly when we train Sim2SG without $\epsilon^{c,pred}$ and evaluate in the same setting as Table 3. Sim2SG with $\epsilon^a + \epsilon^{c,pred}$ gives a 0.246 mAP @0.5 IoU for detection & 0.076 recall@50 for relationship triplets while simply adding $\epsilon^{c,pred}$ to it, we get 0.316 mAP @0.5 IoU for detection & 0.139 recall@50 for relationship triplets (KITTI Hard). This shows the effectiveness of $\epsilon^{c,label}$ and importance of the entire Sim2SG framework.

## 5 CONCLUSION

In this work, we propose Sim2SG, a model that achieves sim-to-real transfer learning for scene graph generation on unlabeled real-world datasets. We decompose the domain gap into label, prediction and appearance discrepancies between synthetic and real domains. We propose methods to address these discrepancies and achieve significant improvements over baselines in all three environments - Clevr, Dining-Sim and Drive-Sim. We do require access to simulator and 3D assets. However, this limitation is mitigated with the availability of open source simulators (Dosovitskiy et al., 2017; Deitke et al., 2020; Kolve et al., 2017; To et al., 2018; Crespi et al., 2020; Denninger et al., 2019; Xiang et al., 2020) and exciting work around object mesh generation (Mescheder et al., 2019; Xu et al., 2019; Gkioxari et al., 2020; Wang et al., 2018). For future work, we plan to learn more complex relationships and explore generative modeling for learning our representation.

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

# A APPENDIX

## A.1 ARCHITECTURE DETAILS

**Encoder $\phi$ and SG Predictor $h$** We use Resnet 101 (He et al., 2016) with imagenet pretraining as the backbone or encoder neural network. We use the Faster-RCNN (Ren et al., 2015) and Graph Convolution Network based architecture from GraphRCNN (Yang et al., 2018) to implement the SG Predictor $h$.

**GRL** For appearance alignment $\epsilon^a$, we use a 2 layer 2D convolution neural network based discriminator with Relu activation. For prediction alignment $\epsilon^{c,pred}$ we use 2 fully connected neural network based discriminator. We also scale the gradients to the encoder network $\phi$ from the discriminator by a factor of 4 in above cases.

## A.2 EXPERIMENTS

### A.2.1 CLEVR

**Setup** The source and target domains of the CLEVR (Johnson et al., 2017) environment leverage Blender (Community, 2018) to render 320x240 images and corresponding ground truth scene graphs. Details of the two domains are available in Section 4.1. We use colors (blue, green, magenta, yellow) and material (metal) for source domain and different colors (pink, brown, white) and material (rubber) for target domain. Additionally, we transform the target by using a style transfer network [1]. For both domains, we sample each class and their size(small, medium & large) with equal probability. The environment has three lights and a fixed camera. We add a small random jitter to their initial positions during the rendering process. Some samples of source and target domain are shown in Figure 4.

**Details of Generation using Pseudo-Statistics** The Label Discrepancy in Section 2.2.1 describes how we generate scenes from pseudo statistics. We assume access to camera parameters.

**Training Details** We run our experiments in two stages. In the first stage, we train with appearance alignment $\epsilon^a$ for 70k iterations. In the second stage, we continue training the model using pseudo statistic based self-learning (label alignment $\epsilon^{c,label}$) for 3 epochs each with 10k iterations.

We optimize the model using a SDG optimizer with learning rate of 0.0001 and momentum of 0.9. We train our model using a batch size 4 on NVIDIA DGX workstations. We report saturation peak performance in all our tables. We give equal regularization weights to source task loss $\epsilon_s$, appearance alignment $\epsilon^a$ and label alignment $\epsilon^{c,label}$.

**Results** More qualitative results of Sim2SG evaluated on the target domain for CLEVR are shown in Figure 5. We see better recall and fewer false positive object detections leading to more accurate scene graphs. Label alignment $\epsilon^{c,label}$ improves object recall, but occasionally introduces some false positive detections. Our appearance alignment $\epsilon^a$ helps in reducing such false positives as shown in Figure 6.

### A.2.2 DINING-SIM

**Setup** The Dining-Sim environment is written using Pixar's USD API and rendered with a proprietary renderer. The source domain is rendered with 2 spp (samples per pixels) followed by denoiser. We select 1 chair (cantilever chair), 1 table (workshop table) and 1 laptop (PC). We randomly place chair and table on the floor and laptop on the floor as well as on the table with a random orientation. The asset for each subcategory is randomly chosen from a list of subcategory specific ShapeNet (Chang et al., 2015) assets. We also ensure that objects do not overlap by applying collision avoidance with simple box collision volumes. A subset of 4 to 5 simple materials that vary only in diffuse colour is created for each of the walls, floor, chair and table. Laptops use the original asset texture.

---

[1] https://github.com/pytorch/examples/tree/master/fast_neural_style

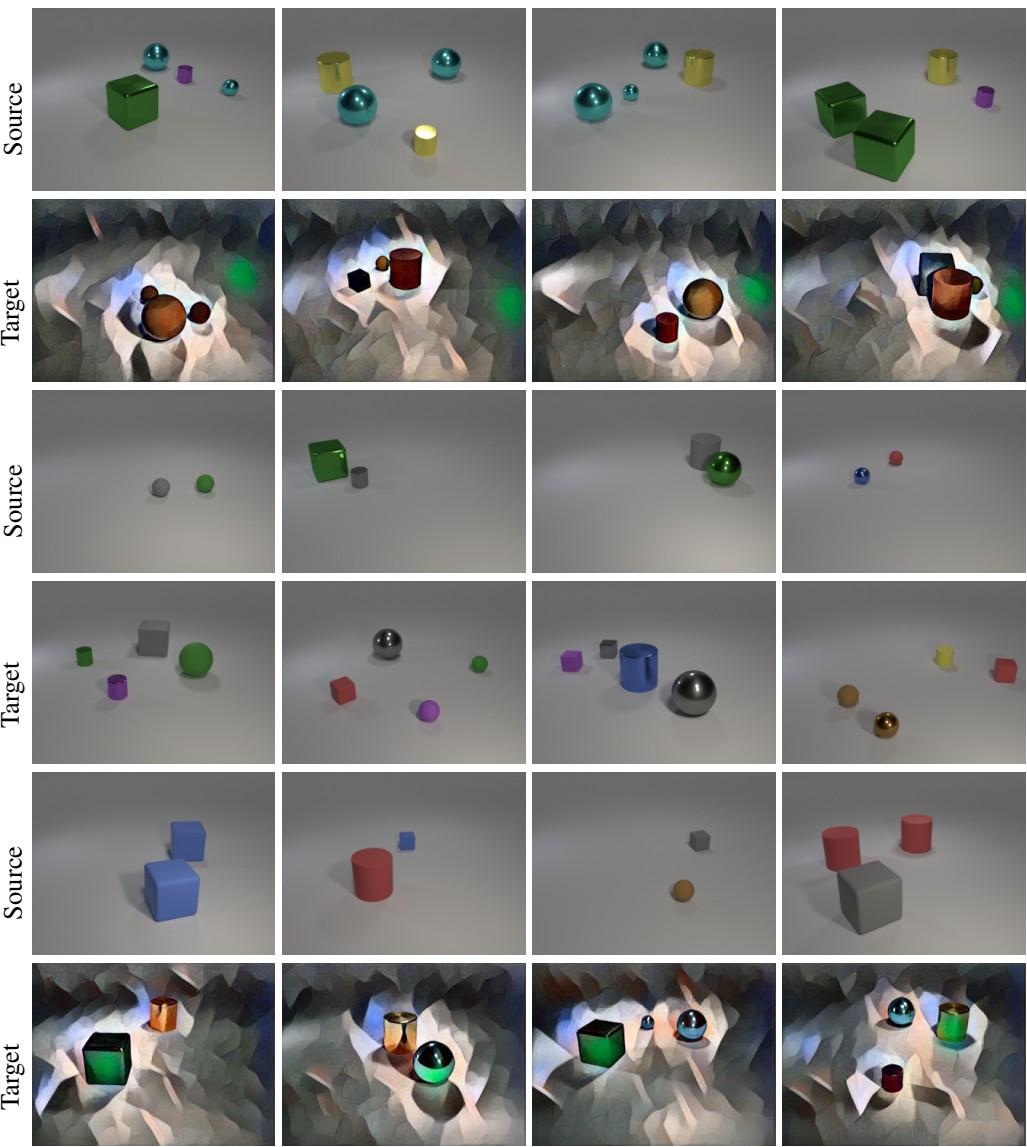

Figure 4: Samples from source and target distributions from Clevr environment. Row 1-2: Source and Target differ in both appearance and content. Row 3-4: Source and Target differ in content but have same appearance. Row 5-6: Source and Target differ in appearance but have same content.

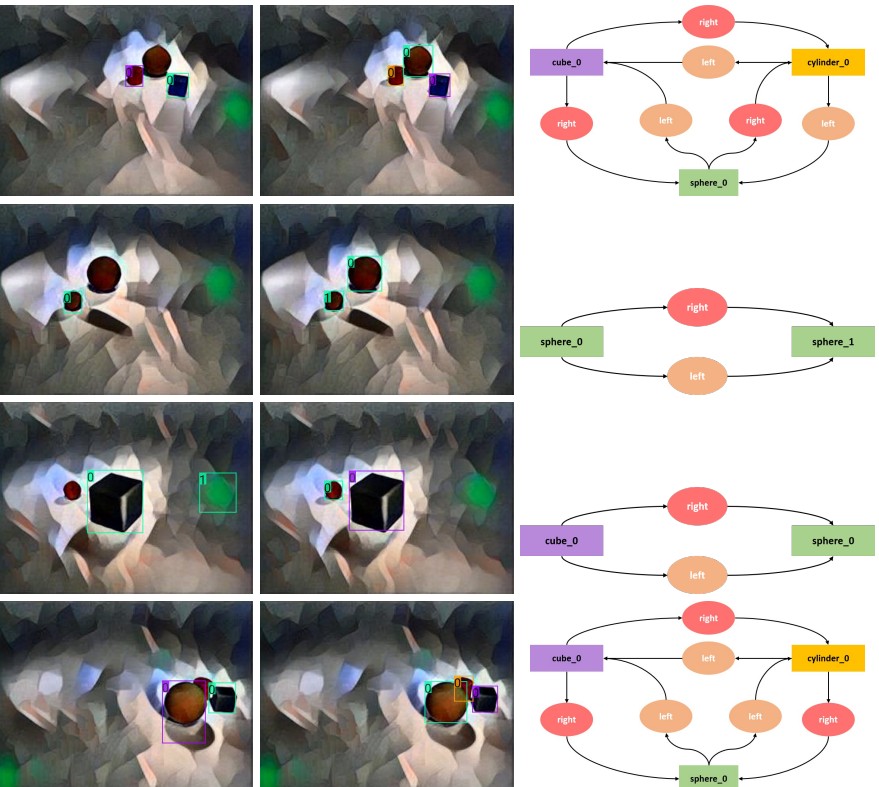

Figure 5: Qualitative results of Sim2SG on the target domain for CLEVR. First column shows that the baseline fails to either detect objects or have high number of false positives (mislabels) leading to poor scene graph. Our method detects objects better, has way fewer false positives and ultimately generates more accurate scene graphs as shown in second and third column respectively. Objects are color coded.

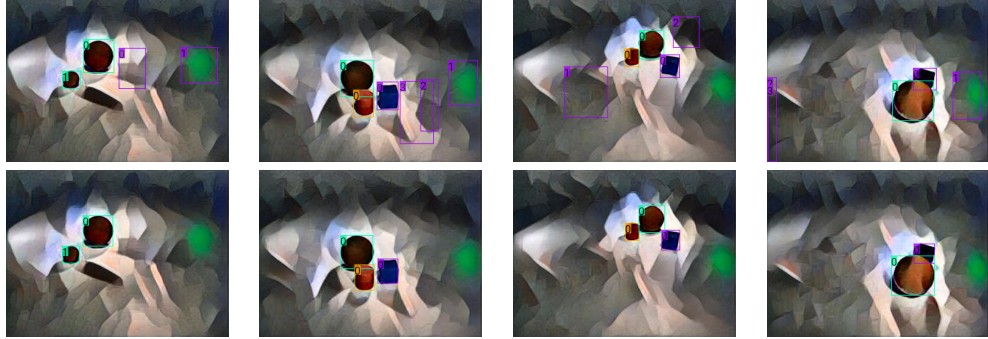

Figure 6: Appearance alignment $\epsilon^a$ reducing false positive. Top row: source + $\epsilon^{c,label}$, bottom row: source + $\epsilon^{c,label}$ + $\epsilon^a$

The target domain is rendered using path tracing with 20 spp (samples per pixels) followed by denoiser. We use 4 chairs (Windsor chair), 1 table (kitchen table) and 2 laptops (MacBook). We first place the table with a random orientation and position on the floor. We then place the four chairs at each side of the table, oriented towards the table centre. Two laptops are then placed randomly on the table surface with a random rotation. The asset for each subcategory is randomly chosen from a list of subcategory specific ShapeNet (Chang et al., 2015) assets. For materials, we use a subset of 4 to 6 physically based, highly detailed materials for each of the walls, floor, chair and table. Laptops use the original asset texture.

Both domains share room parameters: a fixed camera (60 degree field of view, positioned at far side of the room) and 3 fixed spherical lights. Samples from the source and the target domains are shown in Figure 7. There are five kinds of relationships - front, behind, left, right, and on with table as subject. We use 5000 labeled images from source, 5000 unlabeled images from target for training and 1000 labeled images from both source & target domains for evaluation. We use 1024 x 768 image resolution for training and evaluation.

**Details of Generation using Pseudo-Statistics**    Section 2.2.1 describes how we generate synthetic data using pseudo-statistics. We assume access to camera parameters. Pseudo statistic derived from a SG contains a list of objects, their type, 2D centroid position and relationship with others. We filter the objects and relationships among them using an adaptive threshold (details in the next paragraph) for the generation. Using camera parameters, we place each object in the 3D scene by picking a random 3D asset according its type(class) and assigning random pose in the range $0°$–$360°$. We assume context like ground, wall as described in the previous paragraph. We refine the 3D scene further according to the predicted relationships among objects. For example, we use "on" relationship to refine object placements by adjusting the object (laptop or chair) elevation to match the table top.

**Training Details**    We optimize the model using a SGD optimizer with learning rate of 0.0001 and momentum of 0.9. We train our model using a batch size 2 on NVIDIA DGX workstations. We report saturation peak performance in all our tables. We give equal weights to source task loss $\epsilon_s$, appearance alignment $\epsilon^a$, prediction alignment $\epsilon^{c,pred}$ and label alignment $\epsilon^{c,label}$.

We conduct training in three stages. In the first stage, we train for 40,000 iterations on the source domain. Second stage is training the model using pseudo statistic based self-learning (label alignment $\epsilon^{c,label}$) for 6 epochs each with 10k iterations and score threshold of 0.5. We use the aligned synthetic data from this stage (generated with a score threshold of 0.9) to train the next stage. In the third stage, we add appearance alignment $\epsilon^a$ and prediction alignment $\epsilon^{c,pred}$ and train for an additional 20,000 iterations. It takes 24 hours for full training including rendering time.

**Results**    The purpose of the Dining-Sim environment is to show that the Sim2SG works as intended in a more complex setting that is similar to a real-world application. We present the full quantitative results in the table 4 and qualitative results in the Figure 8. We observe that the combination of all alignment terms $\epsilon^{c,label}$, $\epsilon^a$ and $\epsilon^{c,pred}$ gives the best relationship triplet recall of 0.547@50. In order to keep our approach as general as possible, we do not enforce strict rules on object placements and prefer to randomize parameters that are not predicted such as orientation as evident in the qualitative results of label alignment $\epsilon^{c,label}$ in Figure 9. When target domain assets are too dissimilar from the assets in the source domain, it often results in incorrect reconstructions as shown in Figure 9 (last column). We also observe that after label alignment $\epsilon^{c,label}$, the model occasionally has false positive detections, particularly in areas of the floor that have intricate patterns. We qualitatively show that these false positives disappear with the addition of appearance alignment $\epsilon^a$ term (Figure 10).

**Ablations**    We conduct two sets of experiments on the Dining-Sim environment. The first experiment studies appearance gap: source domain has different appearance but similar content from the target domain. The source domain is generated using the target generation scheme but using source dataset materials as shown in Figure 7. We observe that the appearance alignment $\epsilon^a$ helps reduce the appearance gap, increasing relationship triplet recall from 0.625@50 to 0.821@50. Similarly, the second experiment studies content gap where the source and target use the same materials but have different assets, object positions and number of objects. We accomplish this by modifying the source generation scheme to select materials from the target dataset. Samples of source and target are shown in rows 3, 4 of Figure 7. We observe that the label alignment $\epsilon^{c,label}$ term aids in reducing the

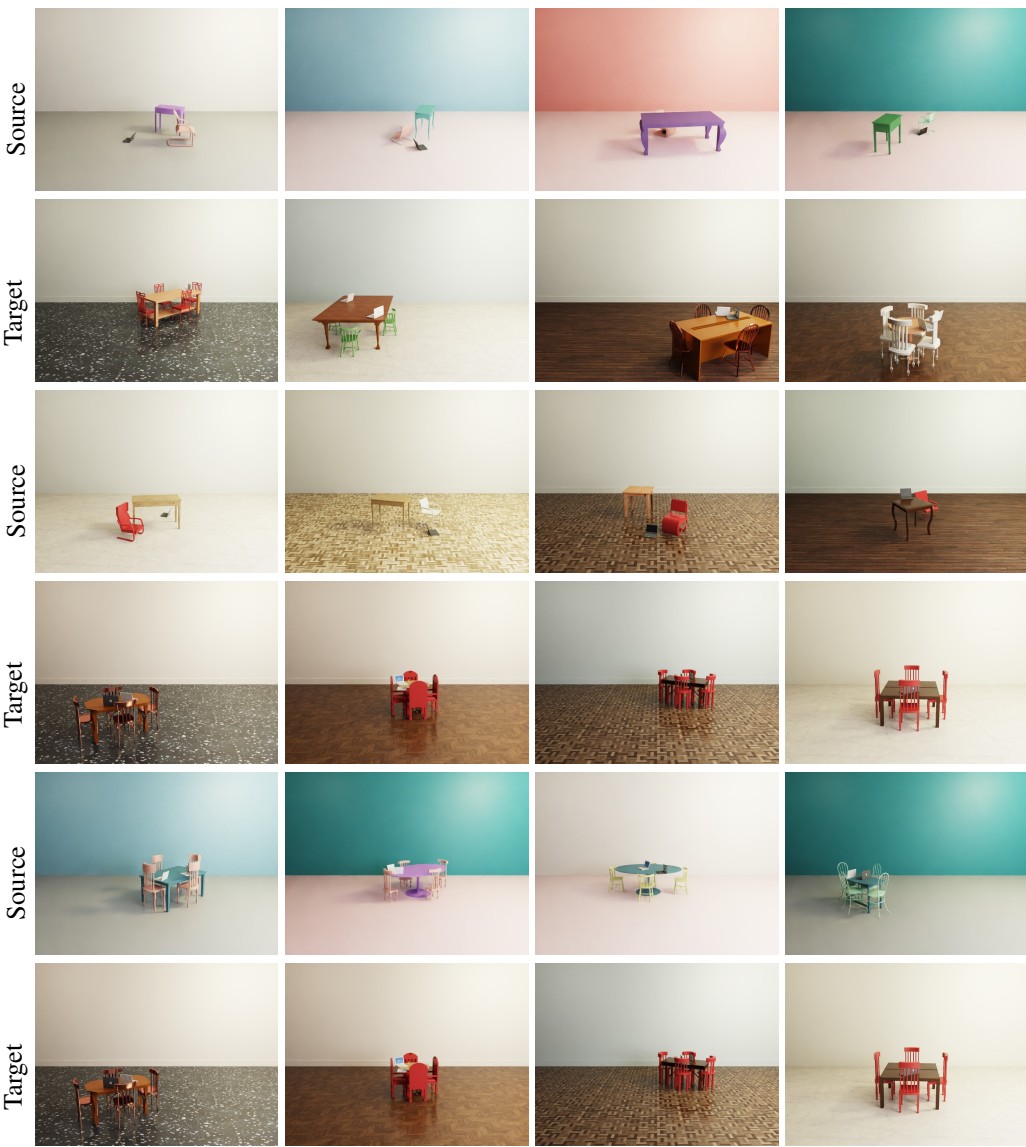

Figure 7: Samples from source and target distributions for Dining-Sim. Row 1-2: Source and Target domains differ in both appearance and content. Row 3-4: Source and Target differ in content but have same appearance. Row 5-6: Source and Target differ in appearance but have same content.

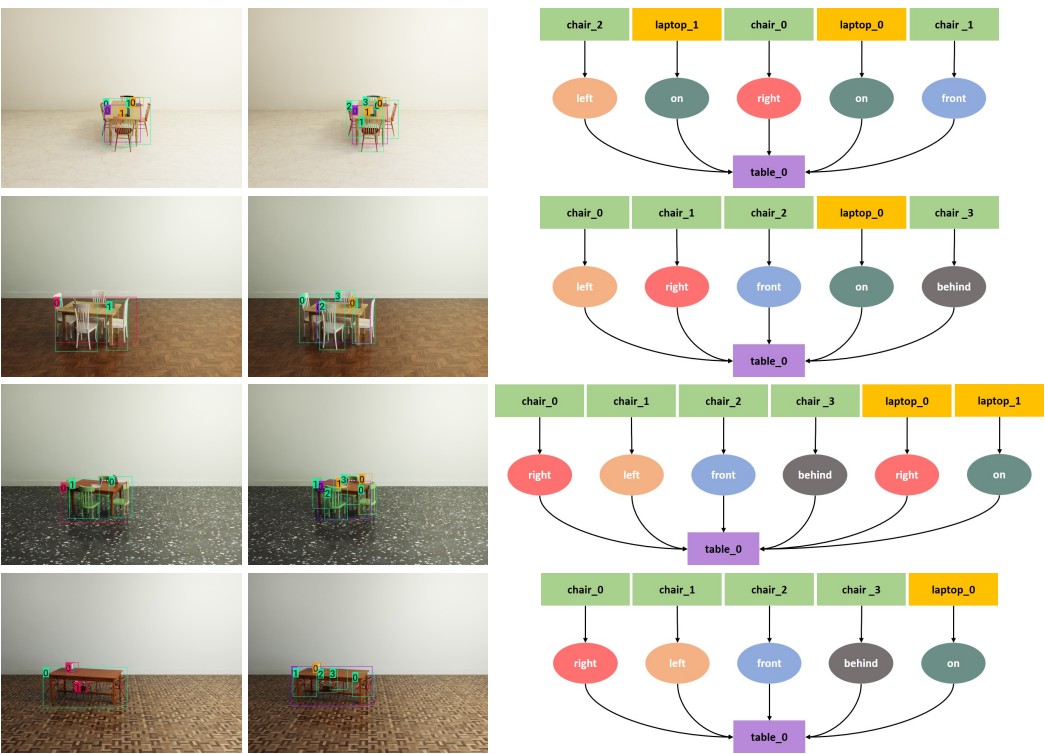

Figure 8: Qualitative results of Sim2SG on the target domain for Dining-Sim. First column shows that the baseline fails to either detect objects or have high number of false positives (mislabels) leading to poor scene graph. Our method detects objects better, has way fewer false positives and ultimately generates more accurate scene graphs as shown in second and third column respectively. Objects are color coded.

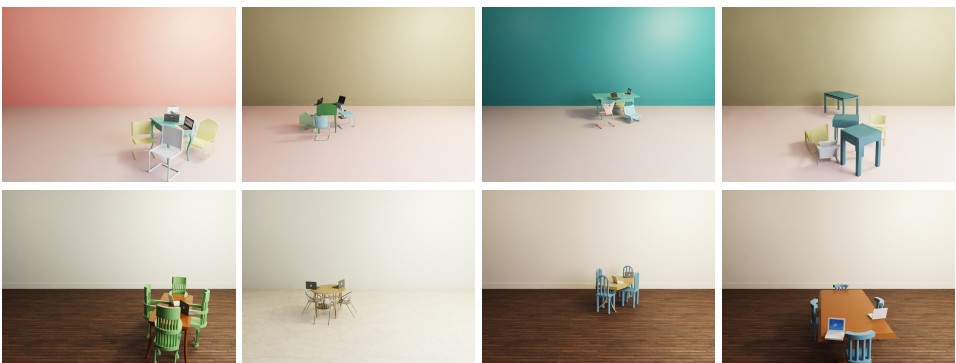

Figure 9: Source reconstructions for target samples in Dining-Sim environment using Label alignment $\epsilon^{c,label}$. Target samples (bottom) and corresponding Source samples (top).

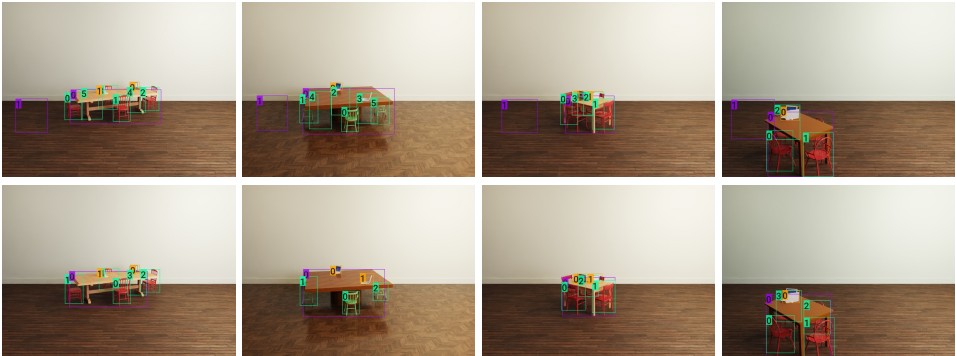

Figure 10: Appearance alignment $\epsilon^a$ reducing false positive. Top row: source + $\epsilon^{c,label}$, bottom row: source + $\epsilon^{c,label}$ + $\epsilon^a$

Table 4: Quantitative results of Sim2SG on a target domain in Dining-Sim environment.

| Trained on | Chair AP | Table AP | Laptop AP | mAP @0.5 IoU | Recall@50 |
|---|---|---|---|---|---|
| Source only | **0.842** ±0.0379 | 0.519 ±0.0881 | 0.392 ±0.0511 | 0.584 ±0.0486 | 0.331 ±0.0637 |
| Ours ($\epsilon^{c,label}$) | 0.737 ±0.0434 | 0.724 ±0.0304 | 0.608 ±0.0467 | 0.713 ±0.0382 | 0.501 ±0.0440 |
| Ours ($\epsilon^{c,label}$ + $\epsilon^a$ + $\epsilon^{c,pred}$) | 0.770 ±0.0220 | **0.757** ±0.0368 | **0.659** ±0.0051 | **0.729** ±0.0147 | **0.547** ±0.0148 |

Table 5: Dining-Sim ablations. Left (resp. right): Source and target domains have different (resp. similar) appearance but similar (resp. different) content distribution. All the evaluations are on the target domain.

| Trained on | mAP @0.5 IoU | Recall@50 | Trained on | mAP @0.5 IoU | Recall@50 |
|---|---|---|---|---|---|
| Source only | $\widehat{0}.772$ ±0.0432 | 0.625 ±0.0763 | Source only | $\widehat{0}.676$ ±0.0112 | 0.468 ±0.0063 |
| Source + $\epsilon^a$ | **0.878** ±0.0006 | **0.821** ±0.0057 | Source + $\epsilon^{c,label}$ | **0.737** ±0.0243 | **0.539** ±0.0061 |

content gap, increasing relationship triplet recall from 0.468@50 to 0.539@50. The relatively modest improvement makes sense as the two domains still differ in content (source and target domain assets differ).

### A.2.3 DRIVE-SIM

**Setup** As mentioned in Section 4.3, we use an Unreal Engine 4[2] based driving simulator akin to (Prakash et al., 2019) to generate synthetic data. We have cars(1-2 per lane), trees(1-3), houses/buildings(1-3), pedestrians(0-2), sidewalk(2), roads(2-6). We do not have poles, street signs or any other objects. We have straight roads. We use realistic random placements, e.g. cars can only be placed on a lane, pedestrians on sidewalk, houses on ground and trees on both sidewalk and ground. We randomize the time of the day, cloud density and use directional light. We assume real world scale. We place our camera at a car height on a random right lane with fixed camera parameters (0 yaw, 0 pitch, 90 fov). We add realistic texture and color to each object similar to (Prakash et al., 2019). We use 1242 x 375 image resolution for training and evaluation.

**Details on Synthetic Data generation using Pseudo-Statistics** Section 2.2.1 describes how we generate synthetic data using pseudo-statistics. We do not have access to KITTI camera parameters and we use the camera parameters described in the previous paragraph. Pseudo statistic derived from a SG contains a list of objects, their type, 2D centroid position and relationship with others. We filter the objects and relationships among them using a confidence threshold of 0.2. Using camera parameters, we place each object in the 3D scene by picking a random 3D asset according its type(class) and assigning random pose in the range $0°–360°$ (except cars that are aligned to the lane). We assume context like road, ground, sky, sidewalk as described in the previous paragraph. We refine the 3D scene further according to the predicted relationships among objects. We also assume a consistent lane width, and number of roads are determined by positions of the detected vehicles in the scene. We place multiple Trees if the projected 3D volume permits.

---

[2]https://www.unrealengine.com/

Table 6: Evaluation on three modes of KITTI : easy (top), moderate (middle), hard (bottom) when training Drive-Sim synthetic environment. The class specific AP and mAP are reported at 0.5 IoU.

| Trained on | Car | Pedes. | House | Veg. | mAP | Recall@50 |
|---|---|---|---|---|---|---|
| Source only (Prakash et al., 2019) | $0.488 \pm 0.007$ | $0.214 \pm 0.025$ | $0.223 \pm 0.022$ | $0.177 \pm 0.010$ | $0.276 \pm 0.005$ | $0.112 \pm 0.009$ |
| DA-FasterRCNN (Chen et al., 2018) | $0.523 \pm 0.036$ | $0.209 \pm 0.038$ | $0.203 \pm 0.037$ | $0.171 \pm 0.012$ | $0.277 \pm 0.017$ | $0.119 \pm 0.022$ |
| Meta-Sim (Kar et al., 2019) | $0.575 \pm 0.008$ | $0.227 \pm 0.024$ | $0.252 \pm 0.008$ | $0.174 \pm 0.033$ | $0.307 \pm 0.003$ | $0.143 \pm 0.007$ |
| Self-learning (Zou et al., 2018) | $0.466 \pm 0.009$ | $0.215 \pm 0.016$ | $0.189 \pm 0.025$ | $0.265 \pm 0.008$ | $0.284 \pm 0.006$ | $0.129 \pm 0.006$ |
| GPA (Xu et al., 2020) | $0.248 \pm 0.053$ | $0.016 \pm 0.026$ | $0.097 \pm 0.030$ | $0.063 \pm 0.031$ | $0.109 \pm 0.022$ | $0.028 \pm 0.009$ |
| SAPNet (Li et al., 2020) | $0.420 \pm 0.052$ | $0.124 \pm 0.035$ | $0.018 \pm 0.004$ | $0.042 \pm 0.010$ | $0.151 \pm 0.010$ | – |
| Ours ($\epsilon^{c,label}$) | $0.566 \pm 0.033$ | $\mathbf{0.310} \pm 0.029$ | $0.261 \pm 0.009$ | $0.242 \pm 0.040$ | $0.345 \pm 0.002$ | $0.193 \pm 0.010$ |
| Ours ($\epsilon^{c,label} + \epsilon^a$) | $0.606 \pm 0.021$ | $0.309 \pm 0.013$ | $0.272 \pm 0.008$ | $0.260 \pm 0.021$ | $0.362 \pm 0.007$ | $0.220 \pm 0.010$ |
| Ours ($\epsilon^{c,label} + \epsilon^a + \epsilon^{c,pred}$) | $\mathbf{0.623} \pm 0.033$ | $0.301 \pm 0.018$ | $\mathbf{0.283} \pm 0.007$ | $\mathbf{0.274} \pm 0.015$ | $\mathbf{0.370} \pm 0.005$ | $\mathbf{0.240} \pm 0.003$ |
| Source only | $0.412 \pm 0.006$ | $0.174 \pm 0.018$ | $0.215 \pm 0.022$ | $0.177 \pm 0.011$ | $0.245 \pm 0.002$ | $0.085 \pm 0.008$ |
| DA-FasterRCNN | $0.472 \pm 0.028$ | $0.181 \pm 0.031$ | $0.203 \pm 0.041$ | $0.168 \pm 0.013$ | $0.256 \pm 0.012$ | $0.091 \pm 0.019$ |
| Meta-Sim | $0.455 \pm 0.040$ | $0.203 \pm 0.026$ | $0.242 \pm 0.009$ | $0.176 \pm 0.029$ | $0.269 \pm 0.007$ | $0.093 \pm 0.005$ |
| Self-learning | $0.377 \pm 0.007$ | $0.174 \pm 0.014$ | $0.197 \pm 0.002$ | $0.263 \pm 0.006$ | $0.253 \pm 0.004$ | $0.077 \pm 0.005$ |
| GPA | $0.201 \pm 0.043$ | $0.016 \pm 0.026$ | $0.106 \pm 0.031$ | $0.065 \pm 0.026$ | $0.096 \pm 0.026$ | $0.018 \pm 0.007$ |
| SAPNet | $0.419 \pm 0.068$ | $0.098 \pm 0.028$ | $0.017 \pm 0.005$ | $0.038 \pm 0.008$ | $0.143 \pm 0.017$ | – |
| Ours ($\epsilon^{c,label}$) | $0.471 \pm 0.033$ | $\mathbf{0.273} \pm 0.027$ | $0.244 \pm 0.010$ | $0.233 \pm 0.036$ | $0.305 \pm 0.006$ | $0.128 \pm 0.008$ |
| Ours ($\epsilon^{c,label} + \epsilon^a$) | $0.511 \pm 0.002$ | $0.266 \pm 0.014$ | $0.251 \pm 0.013$ | $0.256 \pm 0.021$ | $0.321 \pm 0.004$ | $0.155 \pm 0.005$ |
| Ours ($\epsilon^{c,label} + \epsilon^a + \epsilon^{c,pred}$) | $\mathbf{0.529} \pm 0.029$ | $0.249 \pm 0.017$ | $\mathbf{0.262} \pm 0.011$ | $\mathbf{0.270} \pm 0.015$ | $\mathbf{0.328} \pm 0.007$ | $\mathbf{0.170} \pm 0.004$ |
| Source only | $0.382 \pm 0.029$ | $0.168 \pm 0.010$ | $0.211 \pm 0.023$ | $0.174 \pm 0.010$ | $0.234 \pm 0.006$ | $0.070 \pm 0.007$ |
| DA-FasterRCNN | $0.424 \pm 0.028$ | $0.170 \pm 0.029$ | $0.200 \pm 0.041$ | $0.169 \pm 0.014$ | $0.241 \pm 0.014$ | $0.074 \pm 0.015$ |
| Meta-Sim | $0.413 \pm 0.009$ | $0.197 \pm 0.027$ | $0.236 \pm 0.009$ | $0.164 \pm 0.023$ | $0.253 \pm 0.003$ | $0.075 \pm 0.005$ |
| Self-learning | $0.312 \pm 0.006$ | $0.167 \pm 0.006$ | $0.191 \pm 0.003$ | $0.263 \pm 0.006$ | $0.233 \pm 0.004$ | $0.062 \pm 0.003$ |
| GPA | $0.174 \pm 0.040$ | $0.011 \pm 0.016$ | $0.106 \pm 0.031$ | $0.059 \pm 0.027$ | $0.087 \pm 0.020$ | $0.015 \pm 0.005$ |
| SAPNet | $0.362 \pm 0.054$ | $0.085 \pm 0.051$ | $0.116 \pm 0.021$ | $0.067 \pm 0.022$ | $0.157 \pm 0.024$ | – |
| Ours ($\epsilon^{c,label}$) | $0.410 \pm 0.009$ | $\mathbf{0.262} \pm 0.025$ | $0.240 \pm 0.010$ | $0.229 \pm 0.036$ | $0.285 \pm 0.003$ | $0.104 \pm 0.006$ |
| Ours ($\epsilon^{c,label} + \epsilon^a$) | $0.493 \pm 0.004$ | $0.252 \pm 0.014$ | $0.247 \pm 0.012$ | $0.253 \pm 0.020$ | $0.311 \pm 0.311$ | $0.127 \pm 0.004$ |
| Ours ($\epsilon^{c,label} + \epsilon^a + \epsilon^{c,pred}$) | $\mathbf{0.501} \pm 0.006$ | $0.241 \pm 0.018$ | $\mathbf{0.254} \pm 0.010$ | $\mathbf{0.269} \pm 0.014$ | $\mathbf{0.316} \pm 0.004$ | $\mathbf{0.139} \pm 0.004$ |

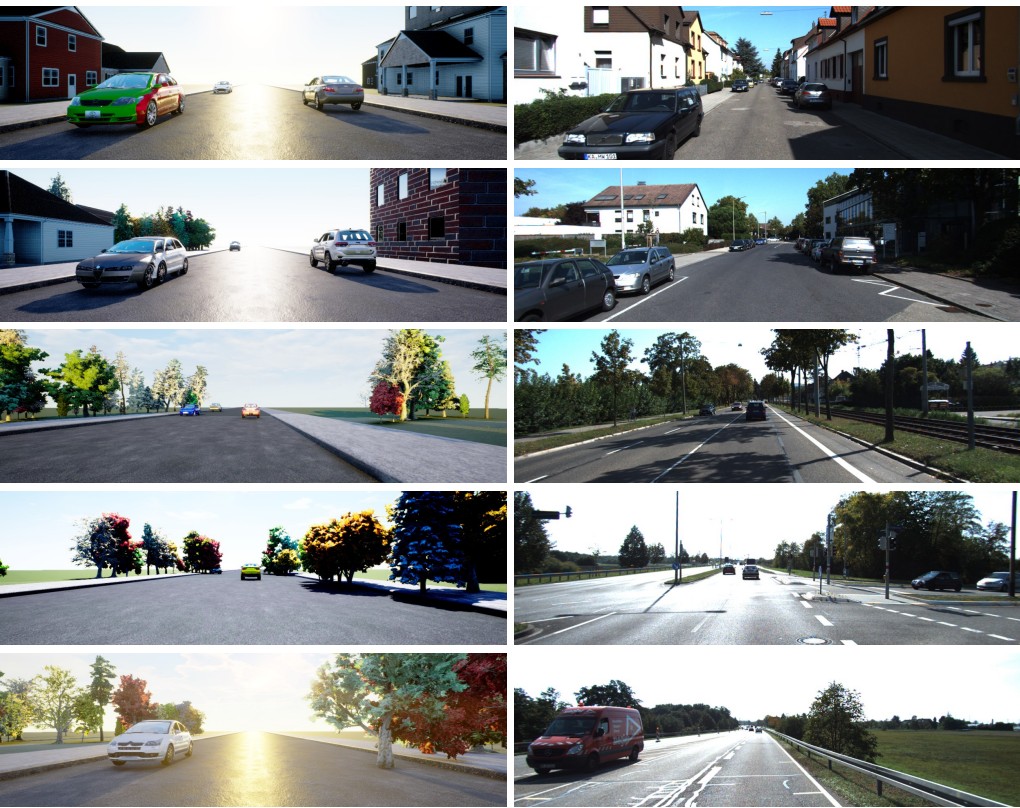

Figure 11: Synthetic reconstructions for target KITTI samples in Drive-Sim environment using Label alignment $\epsilon^{c,label}$. KITTI samples (right) and corresponding synthetic samples (left)

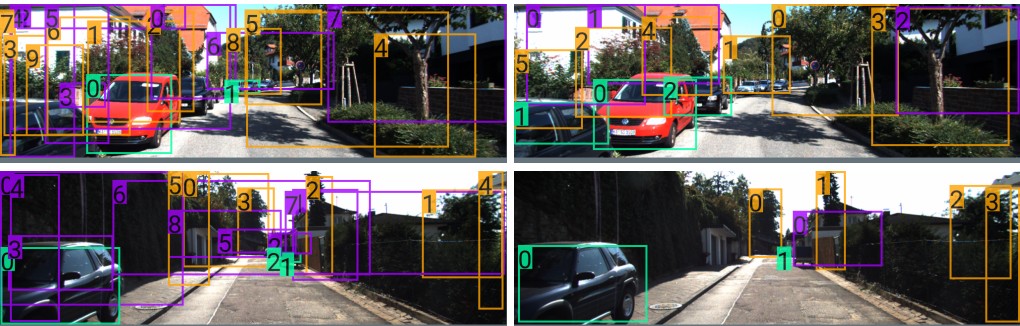

Figure 12: Qualitative results of Sim2SG on the target domain for Drive Sim. First column shows that the source only baseline (Prakash et al., 2019) fails to either detect objects or have high number of false positives (mislabels) leading to poor scene graph. Our method detects objects better, has way fewer false positives as shown in second column. Objects are color coded.

**Training Details**    We optimize the model using a SGD optimizer with learning rate of 0.0001 and momentum of 0.9. We train our model using a batch size 2 on NVIDIA DGX workstations. We report saturation peak performance in all our tables. We give equal regularization weights to source task loss $\epsilon_s$, appearance alignment $\epsilon^a$, prediction alignment $\epsilon^{c,pred}$ and label alignment $\epsilon^{c,label}$.

We run our experiments in two stages. First stage is training the model using pseudo statistic based self-learning (label alignment $\epsilon^{c,label}$) for 3 epochs each with 10k iterations. We use the aligned synthetic data from this stage to train the next stage. During second stage we train the model with appearance alignment $\epsilon^a$ and prediction alignment $\epsilon^{c,pred}$ for an additional 60,000 iterations. This makes sense as $\epsilon^a$ works better when content/labels are aligned between the two domains. The total training takes 24 hours including the rendering time.

**Baselines**:    We adapt domain adaptation baselines (Chen et al., 2018; Xu et al., 2020) to our framework by using the same backbone (Resnet 101) and SG Predictor (GraphRCNN (Yang et al., 2018)) network as Sim2SG, but their loss function. We do not adapt SAPNet (Li et al., 2020). We train these baselines on 6000 images from the source domain (Prakash et al., 2019) using the same optimizer and learning rate as Sim2SG for 60k iterations. We found GPA (Xu et al., 2020) and SAPNet (Li et al., 2020) detection performance to be lower than that reported in their work especially for pedestrian, vegetation and house classes. It is worth noting that their reported class-wise performance numbers only overlap with some of the classes in our work.

We train (Kar et al., 2019) for 40 epochs with a batch size of 16 and learning rate 0.001 as per the authors. We then obtain 6000 images and train it on Sim2SG framework (Resnet 101 backbone and GraphRCNN SG predictor) for 60k iterations using the same optimizer and learning rate as Sim2SG. For self-learning based on pseduo labels (Zou et al., 2018), we obtain the pseudo labels on KITTI images using the most confident predictions by synthetic pretrained GraphRCNN network (as per the authors). We then train these labeled KITTI images on Sim2SG framework for 60k iterations using the same optimizer and learning rate as Sim2SG.

**KITTI Annotation**    We use the existing bounding box annotations of Vehicle and Pedestrians. We annotate Trees and Houses/Buildings of all sizes, occlusion and truncation in KITTI. We use the available camera parameters to project the 2D bounding box into 3D space to help us annotate spatial relationships–front, behind, left and right.

**Results**    Full quantitative evaluations results are in Table 6 on all KITTI (Geiger et al., 2012) evaluation criteria– easy, moderate and hard. In all three criteria, Sim2SG is able to achieve significantly better results (higher detection mAP @0.5 IoU and relationship triplet recall @ 50) than source only baseline (Prakash et al., 2019). More qualitative results of label alignment $\epsilon^{c,label}$ is in Figure 11. We show qualitative improvements (better object recall and fewer false positive detections) over source only (Prakash et al., 2019) baseline in Figure 12 and the corresponding accurate and full scene graphs in Figure 14.

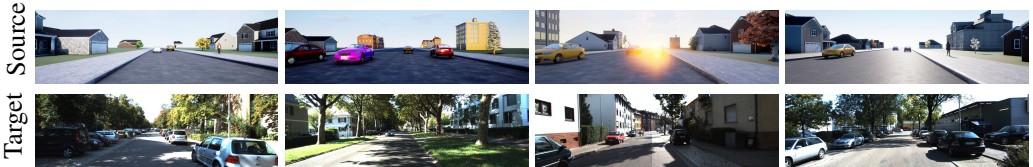

Figure 13: Samples from source and target distributions of Drive-Sim environment including real images from KITTI

Figure 14: Qualitative results of Sim2SG on the target domain for Drive Sim. Sim2SG generates accurate scene graphs.

