# OpenReview forum: "Sim2SG: Sim-to-Real Scene Graph Generation for Transfer Learning"
_ICLR.cc/2021/Conference — Reject_

### Official Review · AnonReviewer3 · 2020-10-25
**This is an interesting work yet (seems to be) have lots of limitations.**

**Rating:** 5
**Confidence:** 3

**Review:**

##########################################################################

Summary:

This paper introduces a framework to utilize the synthetic data as augmentations in the scene graph generation task, which is able to narrow the domain gap by decomposing it into several discrepancies between the two domains. They are the first to propose the synthetic-to-real transfer learning for SGG. The experimental results show the Sim2SG can improve the baseline models in three different scenarios: CLEVR, Dining-Sim, and Drive-Sim.

##########################################################################

Pros:

+ The limitations (long-tailed, noisy, and ambiguous) of the SG dataset have long been noticed in the SGG field. Yet, due to the high costs of collecting the SG dataset, it's hard to provide an ideal dataset for large-scale SGG. This paper introduces a novel idea to augment the SG dataset through synthetic data and transfer learning, which may inspire the later researchers.
+ The proposed framework successfully combines the pseudo-statistic-based self-learning with the Gradient Reversal Layer to solve the intractable label discrepancy and appearance discrepancy.
+ It's able to improve the baseline in different scenarios: CLEVR, Dining-Sim, and Drive-Sim.

##########################################################################

Concerns:

-  (Correct me if I'm wrong) The proposed Sim2SG requires a specifically designed synthetic data generator for each different scenario, e.g., if we want to apply it to the general scene graph dataset like VisualGenome, the generator is also required to be capable of generating all kinds of real-life scenarios. It looks like the proposed method can only be applied to the dataset with limited scenarios.

- There are too many approximations in the implementations of the proposed discrepancies. It's okay but lots of details are missing, which makes the audiences hard to follow, e.g., the details of GRL used in appearance discrepancy and prediction discrepancy.

- The three datasets used in this paper only contains several simple relationships: right, left, front, e.t.c. Does it mean the proposed Sim2SG only works on spatial relationships?


##########################################################################

Reasons for scores:

This paper introduces an interesting direction to augment the scene graph learning, but the proposed method looks like it only works well on a few limited simple scenarios. If I was wrong, please kindly address my concerns above.

##########################################################################

---

> ### Author Response · Authors · 2020-11-14
> **Response to the AnonReviewer3**
>
> * **Applicability to Visual Genome type of Scenarios**. We assume access to simulators for each specific scenario (CLEVR, Dining-Sim and Drive-Sim). The reason is that Sim2SG needs a synthetic data generator to generate label aligned synthetic data. The simulator can be realistic [1,2,6,7] or unstructured/unrealistic (Domain Randomization[3]). With a large collection of assets (e.g. ShapeNet [5]), access to open source simulators [2,6,7] and exciting work around object mesh generation [4,8], we think that this issue is mitigated. As mentioned in our general message to AC, we expect Sim2SG to scale to other scenarios as well because our method (label, prediction and appearance alignment) does not make any domain specific assumptions.
>
> * **Missing details and approximations in Section 2.2**.  We provide the details of Gradient Reversal Layer in Section A.1 and refer to that in Section 2.2.2. We also empirically demonstrate our assumptions in Section 2.2.1 in Section 4.
>
> * **Applicability to other complex scenarios and relationships**. Please see our answer in the main message to AC and reviewers.
>
> References:
>
> [1] Aayush Prakash, Shaad Boochoon, Mark Brophy, David Acuna, Eric Cameracci, Gavriel State, Omer Shapira, and Stan Birchfield. Structured domain randomization: Bridging the reality gap by context-aware synthetic data. In 2019 International Conference on Robotics and Automation (ICRA), pp. 7249–7255. IEEE, 2019.
>
> [2] Alexey Dosovitskiy, German Ros, Felipe Codevilla, Antonio Lopez, and Vladlen Koltun. CARLA: An open urban driving simulator. In CORL, pp. 1–16, 2017.
>
> [3] Jonathan Tremblay, Aayush Prakash, David Acuna, Mark Brophy, Varun Jampani, Cem Anil, Thang To, Eric Cameracci, Shaad Boochoon, Stan Birchfield. Training Deep Networks with Synthetic Data: Bridging the Reality Gap by Domain Randomization. In CVPRW, 2018.
>
> [4] Lars Mescheder, Michael Oechsle, Michael Niemeyer, Sebastian Nowozin, and Andreas Geiger. Occupancy networks: Learning 3d reconstruction in function space. In Proceedings of the IEEE Conference on Computer Vision and Pattern Recognition, pp. 4460–4470, 2019.
>
> [5] Angel X. Chang, Thomas Funkhouser, Leonidas Guibas, Pat Hanrahan, Qixing Huang, Zimo Li, Silvio Savarese, Manolis Savva, Shuran Song, Hao Su, Jianxiong Xiao, Li Yi, and Fisher Yu. Shapenet: An information-rich 3d model repository, 2015.
>
> [6] Eric Kolve, Roozbeh Mottaghi, Winson Han, Eli VanderBilt, Luca Weihs, Alvaro Herrasti, Daniel Gordon, YukeZhu, Abhinav Gupta, and Ali Farhadi.  AI2-THOR: An interactive 3D environment for visual AI.arXivpreprint arXiv:1712.05474, 2017
>
> [7] Adam Crespi, Cesar Romero, Srinivas Annambhotla, Jonathan Hogins, and Alex Thaman. Unity perception,2020.https://blogs.unity3d.com/2020/06/10/.
>
> [8] Georgia Gkioxari, Jitendra Malik, and Justin Johnson. Mesh r-cnn, 2020.

---

### Official Review · AnonReviewer1 · 2020-10-25
**Interesting topic, method technically sound, easy to follow**

**Rating:** 7
**Confidence:** 4

**Review:**

The proposed paper introduces a novel approach for scene graph generation with a focus on bridging the domain gap between synthetic and real data. The proposed model learns the sim-to-real scene graph generation based on labeled synthetic data and unlabeled real data. The method is evaluated on multiple datasets.

Overall, this is an interesting paper that has everything going for it. The topic this work addresses is important from a computer vision, robotics, and computer graphics perspective, the method is technically sound, the results are promising, and the method is evaluated well. Moreover, the paper is well-written and easy to follow. Therefore, I support accepting this work to ICLR.

Comments:

- In Section 2.2 it would help to more carefully define what x and y are.
- In Section 2.2.1 it is not clear why it comes to label shift. It would help to add another sentences describing it.
- I enjoyed reading the discussion provided in Sections 2.2.1 and 2.2.2.
- Currently, the work does not discuss any limitations. As is, it is not clear in which situations the method fails. I would encourage the authors to add a discussion on failure cases to the final version of the paper.
- I would suggest to tone-down the statements on claiming that scene graph labels in sim are mostly free. This claim does not hold as generating scene graphs for applications, such as games, is often a complex endeavor. Once the scene graphs are defined, they can then be used for automatic label generation, but these labels do not come for free. This should be discussed appropriately.
- For completeness sake the related work could also mentioned the following recent paper:

R. Ma, A. Gadi Patil, M. Fisher, M. Li, S. Pirk, B.-S. Hua, S.-K. Yeung, X. Tong, L. Guibas, H. Zhang, Language-Driven Synthesis of 3D Scenes from Scene Databases, ACM Transactions on Graphics (Proceedings of SIGGRAPH Asia), 2018

---

> ### Author Response · Authors · 2020-11-14
> **Response to the AnonReviewer1**
>
> * **Situation where method fails**. Sim2SG can align the label distribution but has problems dealing with class imbalance in the target domain. For example, in the Drive-Sim environment, we believe that pedestrians are under-represented, small and hard to detect class in KITTI. As a result, the Average Precision (AP) of the pedestrian category (Table 3) does not improve with prediction and label alignments. We have added a short discussion about it in Section 4.3.
>
> * **Limitations of Sim2SG**. Please see our answer in the main message to AC and reviewers.
>
> * We have clarified x and y in Section 2.2, explained the label shift in Section 2.2.1 and added the reference “Language-Driven Synthesis of 3D Scenes from Scene Databases” to Section 3.

---

### Official Review · AnonReviewer2 · 2020-10-27
**Complex method, addresses important problem, some concerns on limited empirical evaluation**

**Rating:** 6
**Confidence:** 3

**Review:**

The paper addresses the problem of learning scene graphs from synthetic data and unlabeled real data while performing well on real data by narrowing the content and appearance gap between the two domains when training on synthetic data. Scene graphs are extracted in a two-step process, mapping input to an intermediate latent space and generating the final prediction from the latent space. The authors decompose the content gap into two components: (a) label discrepancy, i.e. how much do the label distributions between the two domains differ, and (b) prediction discrepancy, i.e. the difference in distributions of outputs predicted from the latent space for the two domains. They further model the appearance gap by aligning the latent representation for both domains after accounting for the content gap (to avoid spurious influence of differing content distributions as the latent space is expected to comprise content and appearance). Most of these components are intractable and the paper provides approximations. Empirical investigation on two entirely synthetic and one real/synthetic data set provide evidence for the benefit of the method in closing the posed domain gaps as well as the quality of chosen approximations, the influence of the individual content and appearance gap terms, and the effectiveness of the optimization procedure.

### Strengths:
[S1] The paper addresses a relevant problem: Learning from unlabeled data can be an important component in scaling vision systems to the real-world.

[S2] Claims, contributions and motivations are laid out clearly, evidenced empirically (see W2 below, however) and the writeup is well organized.

[S3] The presented empirical results demonstrate a significant improvement over the state of the art on a challenging problem.

[S4] The method is complex and there are several approximations, but they are largely motivated well and their impact is empirically investigated.

### Weaknesses:
[W1] Although a good amount of detail is provided, due to the complexity of the optimization process, reproducibility of the method seems challenging. I consider this a minor weakness due to the complexity of the task itself.

[W2] Overall, the presented empirical evidence is somewhat limited in complexity: (a) type and number of modeled relations in scene graph, e.g. "car" is always subject for KITTI, (b) number of nodes types: car, pedestrian, vegetation, house. The authors do acknowledge this and put it in context. Nonetheless, it would strengthen the case for the paper if the authors could elaborate more on their expectations when encountering more challenging visual relationship data, either with more relations, more complex relations or more object types. What would be necessary to make the method work there? What part of the method would need to be changed? Would I need to add something to the model?

[W3] I would like to see more discussion on the limits of the proposed approach: What are the expectations on fundamental limitations, e.g. computationally or in terms of representation?

### Further comments
[C1] I may have overlooked this: In table 6 I find it curious that the appearance and prediction discrepancy do not improve results for pedestrians, but only for the other three classes. Is this coincidence or are there speculations on why this is?

### Summary
I believe that methods for efficient transfer learning from synthetic data can have significant impact in a variety of domains in computer vision. As a main weakness of the submission, the presented evidence is largely on toy data or very domain specific ("autonomous driving"), however I do feel that there is sufficient contribution and insight. I suggest to the authors to enable reproducibility of the method by the community to allow more experimentation and further improvements.

---

> ### Author Response · Authors · 2020-11-14
> **Response to the AnonReviewer2**
>
> * **W1: Reproducibility**. We will release the code and KITTI annotations upon acceptance. Sim2SG does require access to a simulator for synthetic domain. However, this limitation is mitigated with the availability of open source simulators [1,2,3,4,5,6,7,8] and exciting work around object mesh generation [9,10,11,12].
>
> * **W2: Applicability to other complex scenarios and relationships**. Please see our answer in the main message to AC and reviewers.
>
> * **W3: Limitations of Sim2SG**. Please see our answer in the main message to AC and reviewers.
>
> * **C1: Pedestrian performance**. We believe that pedestrians are under-represented, small and hard to detect class in KITTI. As a result, the average precision of pedestrian (Table 6) does not improve with prediction and label alignments. We have added a short discussion about it in Section 4.3.
>
> References:
>
> [1] Alexey Dosovitskiy, German Ros, Felipe Codevilla, Antonio Lopez, and Vladlen Koltun. CARLA: An open urban driving simulator. In CORL, pp. 1–16, 2017.
>
> [2] Matt  Deitke,  Winson  Han,  Alvaro  Herrasti,  Aniruddha  Kembhavi,  Eric  Kolve,  Roozbeh  Mottaghi,  JordiSalvador, Dustin Schwenk, Eli VanderBilt, Matthew Wallingford, Luca Weihs, Mark Yatskar, and Ali Farhadi.RoboTHOR: An Open Simulation-to-Real Embodied AI Platform. InCVPR, 2020
>
> [3]  Eric Kolve, Roozbeh Mottaghi, Winson Han, Eli VanderBilt, Luca Weihs, Alvaro Herrasti, Daniel Gordon, YukeZhu, Abhinav Gupta, and Ali Farhadi.  AI2-THOR: An interactive 3D environment for visual AI.arXivpreprint arXiv:1712.05474, 2017
>
> [4]  Thang To, Jonathan Tremblay, Duncan McKay, Yukie Yamaguchi, Kirby Leung, Adrian Balanon, Jia Cheng,and Stan Birchfield. NDDS: NVIDIA deep learning dataset synthesizer, 2018.https://github.com/NVIDIA/Dataset_Synthesizer.
>
> [5] Adam Crespi, Cesar Romero, Srinivas Annambhotla, Jonathan Hogins, and Alex Thaman. Unity perception,2020.https://blogs.unity3d.com/2020/06/10/.
>
> [6] Maximilian Denninger, Martin Sundermeyer, Dominik Winkelbauer, Youssef Zidan, Dmitry Olefir, Mohamad Elbadrawy, Ahsan Lodhi, and Harinandan Katam. Blenderproc.arXiv preprint arXiv:1911.01911, 2019.
>
> [7] Fanbo Xiang, Yuzhe Qin, Kaichun Mo, Yikuan Xia, Hao Zhu, Fangchen Liu, Minghua Liu, Hanxiao Jiang,Yifu Yuan, He Wang, et al. Sapien: A simulated part-based interactive environment. InProceedings of theIEEE/CVF Conference on Computer Vision and Pattern Recognition, pp. 11097–11107, 2020.
>
> [8] Nimier-David, Merlin and Vicini, Delio and Zeltner, Tizian and Jakob, Wenzel. Mitsuba 2: A Retargetable Forward and Inverse Renderer. Association for Computing Machinery, 2019.
>
> [9] Lars Mescheder, Michael Oechsle, Michael Niemeyer, Sebastian Nowozin, and Andreas Geiger. Occupancy networks: Learning 3d reconstruction in function space. In Proceedings of the IEEE Conference on Computer Vision and Pattern Recognition, pp. 4460–4470, 2019.
>
> [10] Qiangeng Xu, Weiyue Wang, Duygu Ceylan, Radomir Mech, and Ulrich Neumann. Disn: Deep implicit surfacenetwork for high-quality single-view 3d reconstruction, 2019.
>
> [11] Georgia Gkioxari, Jitendra Malik, and Justin Johnson. Mesh r-cnn, 2020.
>
> [12] Nanyang Wang, Yinda Zhang, Zhuwen Li, Yanwei Fu, Wei Liu, and Yu-Gang Jiang. Pixel2mesh: Generating 3dmesh models from single rgb images, 2018.

---

### Official Review · AnonReviewer4 · 2020-10-30

**Rating:** 5
**Confidence:** 3

**Review:**

## Summary
The paper tackles the problem of sim2real transfer for scene graph inference. It proposes an approach for closing the gap between simulated training data and real test data, to allow models trained purely on simulated data to be deployed on real images. The approach is tested on multiple environments, including transfer from a driving scene simulator to real KITTI scenes.

## Strengths
- the problem of sim2real transfer is important, especially for training scene graph inference models, for which it is very expensive to obtain ground-truth scene graph supervision
- the paper covers a wide range of related works on scene graph inference and domain adaptation
- the approach is compared to a range of baseline approaches and improves performance on transfer to KITTI images
- the paper ablates several components of the proposed approach and shows qualitative prediction results

## Weaknesses
- **unclear writing**: the writing of introduction and approach section does not clearly express the novel idea presented in the paper. After reading over the paper multiple times, I now understand that the main novelty lies in the usage of pseudo-statistics of the target domain to adjust the source domain data distribution, which is possible since the paper is looking at *sim2real transfer* as a special case of domain transfer in which we have full control of the source domain. In contrast, most of the derivation in section 2.2 was already presented in Wu et al. 2019, except that they assumed to have no control over the label distribution (since they assumed the general domain transfer setting where the source data distribution is fixed too). Based on this, I think the main contribution of the paper should be explained much more clearly (see suggestions below).
- **unclear focus**: the paper claims two seemingly orthogonal contributions: (1) applying sim2real transfer to the problem of scene graph inference and, (2) introducing an approach for modifying the training data distribution to better match the "real" test data distribution specifically for sim2real domain adaptation scenarios. Both problems are valid on their own (even though (1) might not be a sufficient contribution by itself), but the paper fails to make a coherent argument why both should be jointly investigated. Focussing on one of the problems can improve the clarity of the paper.
- **training data alignment requires domain-specific knowledge**: the core novelty proposed in the paper is the usage of pseudo-statistics to adjust the source data distribution. While the main paper remains unclear *how exactly* the pseudo-statistics are translated to a new training dataset, the appendix has more details that describe how the process is specifically adjusted to every domain. This domain-specific adjustment (like randomly placing trees in the "permittable" region in the KITTI training scenes) represents additional domain knowledge which conventional sim2real approaches do not require
- **baselines not explained in detail**: the experimental section does not properly explain the features of the baselines that the approach is compared to. Without a more detailed explanation it is hard to follow the discussion in the "Results" paragraph of section 4.3 that discusses the worse baseline performance. Further, the mentioned dataset sizes in the experimental section are confusing since the proposed method is continually re-generating its synthetic training data which makes it unclear whether the comparison to baselines operating on a fixed dataset is fair.
- **no analysis experiments**: the experimental section lacks experiments/visualizations that give a more detailed analysis of the novel part of the proposed algorithm. Specifically, it would be nice to see how the approaches changes the training data distribution over the course of training or to analyze the stability of the iterated procedure of training --> data adjustment --> training...
- **details about derivation unclear**: it seems that crucial parts of the derivation of equation (2) were moved to the appendix without referring to them in the text, which makes it very hard to understand section 2.2, which is the core section for the proposed approach

## Questions
- the reduction of the prediction discrepancy aims to reduce the gap between the predicted label distributions, but the approach is manually aligning these distributions (by computing the pseudo-statistics), so training the model to accurately fit the training data label distribution should already minimize this discrepancy, so why do we need an additional loss for that? (it anyways seems to be only used in one out of the three experiments, why only there?)
- the experimental section mentions fixed sizes for the used synthetic datasets, however, the proposed method is iteratively generating new data every epoch -- how is that accounted for in the dataset size? do the baselines get access to comparable amounts of synthetic data?

## Suggestions to improve the paper
- determine a clear focus for the paper: the most coherent story would be to focus on domain adaptation for sim2real transfer using pseudo-label source content alignment, but then additional experiments on non-scene-graph-inference problems would be required to show that the method is generally applicable.
- improve writing clarity by clearly mentioning how the investigated problem differentiates from the more general domain adaptation problem, then show (best experimentally) how usual domain adaptation methods struggle with the content gap between source and target environment and how the ability to modify the training distribution in the sim2real setting leads to the proposed method
- expand on the explanation of the baselines (+test them in all environments) to make it clearer why they fail (particularly since there is no reference work for the sim2real scene graph inference problem)
- add experiments that analyze the technically novel part of your paper, show how the training distribution changes over time to approach the target distribution
- if possible, add a baseline that was trained with target, real image labels, particularly in KITTI environment, to show the performance gap of sim2real methods
- the proposed approach seems like an EM-like procedure: compute pseudo-statistics with the current model + adjust synthetic training data distribution (E-step), then train model on the new data (M-step). Presenting it in this way could make the procedure clearer. It would also be good to discuss potential tradeoffs with prior work (eg is the EM-like procedure prone to instability, do we need any additional assumptions like which quantities to compute pseudo-statistics over and which to ignore?)
- the caption of table 2 does not mention for which dataset the evaluation is performed (it is mentioned in the text, but it would be good to add it to the caption too)

## Overall Recommendation
Overall, the lack of clarity in the paper's writing makes it hard to clearly grasp the core problem and delta to prior work, but I believe that some major restructuring and a few additional analysis experiments can substantially improve the clarity and focus of the paper. In its current form it is not ready for acceptance.

---

> ### Author Response · Authors · 2020-11-14
> **Response to the AnonReviewer4**
>
>
> * **Unclear writing**. We thank the reviewer for pointing out some ambiguities in expressing the novel idea of our approach. Sim2SG differs from other unsupervised domain adaptation methods [1,2,3]  as it can modify the source distribution (via self-learning based on pseudo-statistics to align with the target distribution) with access to a synthetic data generator. We also outperform these domain adaptation baselines [1,2,3]  as shown in Section 4.3. We have added this to Section 1.
>
> * **Unclear focus**. The focus of the paper is sim2real transfer for the scene graph generation task. However, Scene Graphs (SGs) do not work well with synthetic data simulators [4] because of domain gap. We explain this problem in Section 1 (paragraph 3).  Hence we propose to minimize the label (ground truth), prediction and appearance discrepancies between the two domains using pseudo-statistic based self-learning and adversarial techniques. (Section 1, fourth and fifth paragraph).
> The two contributions that you mention are not orthogonal since we propose the second contribution as a way to solve the first contribution.
>
> * **Domain-specific knowledge for training**. We assume access to domain specific simulators to generate synthetic data whose labels are aligned with real data. These simulators can be unstructured  (this is the case for CLEVR, Dining-Sim) or structured/realistic (Driving-Sim). While access to structured/realistic simulators [4,5,6] may  lead  to better context (road, Sky, sidewalk,etc) and  performance, we have shown that Sim2SG works well with unstructured simulators (Dining-Sim) as well.
> The fact that we tried more or less realistic generators in diverse contexts can be seen as an ablation study of our approach.
>
> * **Baselines**. We have added a functional description of the baselines in the second paragraph of Section 4.3. The hyperparameters are mentioned in Section A.2.3. The final numbers reported in Table 3 from Sim2SG are on fixed data obtained after 3 epochs of label alignment as well. This is similar to Meta-Sim that was also trained for several epochs to generate final fixed data of the same size.  We use fixed data and exact same size for all other baselines as well.
>
> * **Analysis Experiments**. We thank the reviewer for this excellent suggestion. Next week, we will add a qualitative analysis of synthetic data towards label alignment as shown in Figure 10. We will send a new message when the revised manuscript includes the suggested analysis.
>
> * **Missing details of Equation 2**. We moved the derivation from Appendix to Equation 2 in the revised version.
>
> * **Analogy to  EM-like procedure**. We thank the reviewer for this suggestion. Our approach is indeed similar to EM in the sense that the model parameters of the SG generator are exploited to improve the parameters of the discriminator and vice versa. We have added a line to illustrate the analogy in Section 2.2.1.
>
> * **Additional loss for label discrepancy**.  Label discrepancy is one of the three discrepancies that we address in Sim2SG. Even though we do not minimize the label discrepancy directly, it represents a bound and helps to automatically align the label distribution (via pseudo statistic and synthetic generator). Sim2SG shows improvement over baselines using this term and without it, appearance and label alignment fail to improve the performance (ablation in Section 4.3).
>
> * **Oracle performance on KITTI**. We do not have access to all the labels in KITTI, so we cannot report the oracle performance. However we have added the oracle performance for Dining Sim environment in Section 4.2
>
> References:
>
> [1] Yuhua Chen, Wen Li, Christos Sakaridis, Dengxin Dai, and Luc Van Gool. Domain adaptive faster r-cnn for object detection in the wild. 2018 IEEE/CVF Conference on Computer Vision and Pattern Recognition, Jun 2018.
>
> [2] Minghao Xu, Hang Wang, Bingbing Ni, Qi Tian, and Wenjun Zhang. Cross-domain detection via graph-induced prototype alignment. 2020 IEEE/CVF Conference on Computer Vision and Pattern Recognition (CVPR), Jun 2020
>
> [3] Congcong Li, Dawei Du, Libo Zhang, Longyin Wen, Tiejian Luo, Yanjun Wu, and Pengfei Zhu. Spatial attention pyramid network for unsupervised domain adaptation, 2020.
>
> [4] Aayush Prakash, Shaad Boochoon, Mark Brophy, David Acuna, Eric Cameracci, Gavriel State, Omer Shapira, and Stan Birchfield. Structured domain randomization: Bridging the reality gap by context-aware synthetic data. In 2019 International Conference on Robotics and Automation (ICRA), pp. 7249–7255. IEEE, 2019.
>
> [5] Eric Kolve, Roozbeh Mottaghi, Winson Han, Eli VanderBilt, Luca Weihs, Alvaro Herrasti, Daniel Gordon, YukeZhu, Abhinav Gupta, and Ali Farhadi.  AI2-THOR: An interactive 3D environment for visual AI.arXivpreprint arXiv:1712.05474, 2017
>
> [6] Adam Crespi, Cesar Romero, Srinivas Annambhotla, Jonathan Hogins, and Alex Thaman. Unity perception,2020.https://blogs.unity3d.com/2020/06/10/.

---

> > ### Author Response · Authors · 2020-11-17
> > **Response to the AnonReviewer4:**
> >
> > * **Analysis Experiments**. We have added Figure 3 to Section 4.3 (second last paragraph) that illustrates the evolution of the generated data through different epochs of the training process using label alignment.

---

### Author Response · Authors · 2020-11-14
**Response to all the Reviewers and the Area Chair**

We thank the reviewers for their positive feedback and valuable suggestions. In our paper, we propose Sim2SG: a technique for sim-to-real transfer for scene graph (SG) generation with key novelty around sim2real content gap. Sim2SG decomposes the domain gap based on label, prediction and appearance discrepancies between the synthetic (source) and unlabeled real (target) domains. The strengths of the work are as follows:

* Leveraging simulated data to improve performance on real-world scene graphs is an important problem (R4, R1, R3) and learning from unlabeled data can be an important component in scaling vision systems to the real-world. (R2)
* The approach is compared to a range of baseline approaches and improves performance on a challenging problem (R4, R2, R3)
* Addressing content gap between the two domains (R4)

Some common concerns of the reviewers include:

* **Applicability to other complex scenarios and relationships (R2, R3)**. We have experimentally shown that Sim2SG works in three different and diverse scenarios -- CLEVR, Dining-Sim and Driving Sim. With access to the simulator we can create scenarios with different kinds of relationships which can be actional (e.g. running), prepositional (with), comparative (taller).
We can then train Sim2SG with the simulator and unlabeled real data with these complex relationships. We expect Sim2SG to work with these relationships as they are a function of localization of objects and/or their features. These are similar to the spatial relationships we exploit in the paper. Indeed, Sim2SG addresses the domain gap by minimizing the label, prediction and appearance discrepancy of the two domains without any domain specific assumptions.  For this reason, we expect Sim2SG to scale with more objects and types as well.

* **Limitations of Sim2SG (R2, R1)**. We briefly discussed in Section 5 the limitation  about requiring domain specific simulators and expensive 3D assets.  However, this limitation is mitigated with the availability of open source simulators [1,2,3,4,5,6,7,8] and exciting work around object mesh generation [9,10,11,12].

References:

[1] Alexey Dosovitskiy, German Ros, Felipe Codevilla, Antonio Lopez, and Vladlen Koltun. CARLA: An open urban driving simulator. In CORL, pp. 1–16, 2017.

[2] Matt  Deitke,  Winson  Han,  Alvaro  Herrasti,  Aniruddha  Kembhavi,  Eric  Kolve,  Roozbeh  Mottaghi,  JordiSalvador, Dustin Schwenk, Eli VanderBilt, Matthew Wallingford, Luca Weihs, Mark Yatskar, and Ali Farhadi.RoboTHOR: An Open Simulation-to-Real Embodied AI Platform. InCVPR, 2020

[3]  Eric Kolve, Roozbeh Mottaghi, Winson Han, Eli VanderBilt, Luca Weihs, Alvaro Herrasti, Daniel Gordon, YukeZhu, Abhinav Gupta, and Ali Farhadi.  AI2-THOR: An interactive 3D environment for visual AI.arXivpreprint arXiv:1712.05474, 2017

[4]  Thang To, Jonathan Tremblay, Duncan McKay, Yukie Yamaguchi, Kirby Leung, Adrian Balanon, Jia Cheng,and Stan Birchfield. NDDS: NVIDIA deep learning dataset synthesizer, 2018.https://github.com/NVIDIA/Dataset_Synthesizer.

[5] Adam Crespi, Cesar Romero, Srinivas Annambhotla, Jonathan Hogins, and Alex Thaman. Unity perception,2020.https://blogs.unity3d.com/2020/06/10/.

[6] Maximilian Denninger, Martin Sundermeyer, Dominik Winkelbauer, Youssef Zidan, Dmitry Olefir, Mohamad Elbadrawy, Ahsan Lodhi, and Harinandan Katam. Blenderproc.arXiv preprint arXiv:1911.01911, 2019.

[7] Fanbo Xiang, Yuzhe Qin, Kaichun Mo, Yikuan Xia, Hao Zhu, Fangchen Liu, Minghua Liu, Hanxiao Jiang,Yifu Yuan, He Wang, et al. Sapien: A simulated part-based interactive environment. InProceedings of theIEEE/CVF Conference on Computer Vision and Pattern Recognition, pp. 11097–11107, 2020.

[8] Nimier-David, Merlin and Vicini, Delio and Zeltner, Tizian and Jakob, Wenzel. Mitsuba 2: A Retargetable Forward and Inverse Renderer. Association for Computing Machinery, 2019.

[9] Lars Mescheder, Michael Oechsle, Michael Niemeyer, Sebastian Nowozin, and Andreas Geiger. Occupancy networks: Learning 3d reconstruction in function space. In Proceedings of the IEEE Conference on Computer Vision and Pattern Recognition, pp. 4460–4470, 2019.

[10] Qiangeng Xu, Weiyue Wang, Duygu Ceylan, Radomir Mech, and Ulrich Neumann. Disn: Deep implicit surfacenetwork for high-quality single-view 3d reconstruction, 2019.

[11] Georgia Gkioxari, Jitendra Malik, and Justin Johnson. Mesh r-cnn, 2020.

[12] Nanyang Wang, Yinda Zhang, Zhuwen Li, Yanwei Fu, Wei Liu, and Yu-Gang Jiang. Pixel2mesh: Generating 3dmesh models from single rgb images, 2018.

---

### Decision · Program_Chairs · 2021-01-07
**Final Decision**

**Decision:**

Reject

**Comment:**

Paper addresses the problem of sim2real (training with synthetic data and then applying the  learned model on real data) in the context of scene graph generation.  The paper was reviewed by four expert reviewers  who identified the following pros and cons of  the method.

> Pros:
- Paper addresses relevant and important problem [R1, R2, R4]
- Paper containing compelling results with respect to a number of baselines [R2, R4]

> Cons:
- Lack of clear motivation, focus, and explanation of novelty [R4]
- Missing details, which makes paper hard to follow [R3, R4]
- Lack of explanations for baselines [R4]
- Lack of focused analysis of specific contributions [R4]
- Lack of discussion on the limitations of the approach [R1, R2]
- Presented evidence is largely on toy data or very domain specific [R2]

A number of the shortcoming were addressed by the authors during the rebuttal through revisions.  However, opinion of reviewers on the paper remained split, with paper receiving  following scores:

- 5: Marginally below acceptance threshold
- 6: Marginally above acceptance threshold
- 7: Good paper, accept
- 5: Marginally below acceptance threshold

Overall, all reviewers and AC agree that the paper addresses an important and interesting problem. At the same time  AC agrees with R2 and others that point out that there are significant limitation in terms of applicability of the approach in more complex scenarios, where readily available simulator may not exist. On balance, and considering the large number of high quality submissions to ICLR this year, the paper was deemed marginally below the acceptance threshold.